# *Memba*: Membrane-driven Parameter-Efficient Fine-Tuning for Mamba

**Donghyun Lee**[1]  **Yuhang Li**[2]  **Ruokai Yin**[2]  **Shiting Xiao**[2]  **Priyadarshini Panda**[1]

[1]Electrical and Computer Engineering, University of Southern California
[2]Electrical Engineering, Yale University
`{donghyun.lee.1}@usc.edu`

## Abstract

State Space Models (SSMs) have emerged as powerful alternatives to attention-based Transformers, with Mamba demonstrating impressive efficiency and scalability. As these models grow increasingly larger, the need for Parameter-Efficient Fine-Tuning (PEFT) methods becomes critical to adapt pre-trained Mamba to downstream tasks without prohibitive computational costs. However, previous approaches simply apply traditional Transformer-tailored PEFT methods without addressing the unique temporal processing dynamics of SSMs. To address this limitation, we propose *Memba*, a membrane-driven PEFT approach specifically designed for Mamba. *Memba* introduces Leaky Integrate Membrane (LIM) neurons as bio-inspired gating mechanisms that naturally accumulate membrane potentials over time, enhancing selective information retention. By strategically combining LIM neurons with Low-Rank Adaptations (LoRA) and cross-layer membrane transfer, our approach significantly improves Mamba's temporal modeling capabilities. Extensive experiments across language and vision tasks demonstrate that *Memba* achieves substantial improvements over existing PEFT methods. The code is available at https://github.com/Intelligent-Computing-Lab-Panda/Memba.

## 1 Introduction

State Space Models (SSMs) (Gu et al., 2021b;a; Fu et al., 2022) have emerged as powerful alternatives to Transformer (Vaswani et al., 2017) architectures, offering linear computational complexity with respect to sequence length while maintaining competitive performance. SSMs share functional similarities with recurrent architectures, including Long Short-Term Memory (LSTM) (Hochreiter & Schmidhuber, 1997) and Gated Recurrent Unit (GRU) (Chung et al., 2014), through evolving hidden states, though they employ different mathematical foundations based on state space theory (Gu et al., 2020). Recent advancements, particularly Mamba (Gu & Dao, 2023; Dao & Gu, 2024), have demonstrated remarkable success across language modeling (Pióro et al., 2024; Wang et al., 2024), computer vision (Liu et al., 2024c; Zhu et al., 2024), and other domains (Wang et al., 2025; Quan & Li, 2024; Ota, 2024; Hu et al., 2024) by introducing selective SSMs with data-dependent parameters. As these models scale, Parameter-Efficient Fine-Tuning (PEFT) methods become crucial for adaptation with minimal trainable parameters. While PEFT techniques have shown success in Transformer-based models (Hu et al., 2022; Houlsby et al., 2019), their application to SSMs remains limited. Recent works (Yoshimura et al., 2024; Halloran et al., 2024; Ham et al., 2024) have begun exploring PEFT for Mamba, but simply transfer Transformer-tailored methods without addressing the unique temporal processing dynamics of SSMs.

Although Mamba is meticulously designed based on state-space theory (Gu et al., 2020), current architecture lacks the sophisticated gating structures found in traditional recurrent networks such as LSTM (Hochreiter & Schmidhuber, 1997) and GRU (Chung et al., 2014), relying instead on a single linear transformation. Traditional recurrent networks incorporate multiple trainable gates to manage memory retention and forgetting over time. In contrast, Mamba's simplified gating mechanism lacks temporal selectivity, structured memory, and nonlinear control capabilities. We believe that this limitation, shared with earlier SSMs such as S4 (Gu et al., 2021a), can hinder the model's ability to adaptively capture task-specific temporal information during fine-tuning. Furthermore,

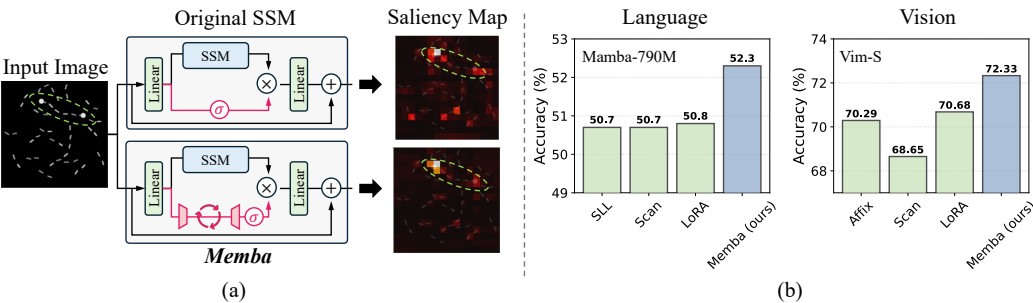

Figure 1: Overview of ***Memba*** architecture and performance comparison. (a) Architecture and saliency map comparison between original SSM and ***Memba*** on a Pathfinder dataset image. The pink lines in architectures represent gating branches, and the green dashed circle indicates the target path to be identified. (b) Performance comparison on language (commonsense reasoning) and vision (VTAB-1k) tasks using Mamba-790M and Vim-S architectures respectively. We compare ***Memba*** with SLL LoRA, Additional-scan, Affix-tuning, and LoRA in (Yoshimura et al., 2024).

recent studies (Yoshimura et al., 2024; Ham et al., 2024) have revealed that directly fine-tuning the state-space components often degrades Mamba's performance, suggesting that modifying temporal processing mechanisms during adaptation is particularly challenging. This phenomenon raises a fundamental question: ***how can we effectively incorporate temporal adaptation during fine-tuning without disrupting the balanced dynamics of pre-trained SSMs?***

In this work, we propose ***Memba***, which introduces a membrane-driven PEFT technique to enhance SSM gating. At the core of our approach is the Leaky Integrate Membrane (LIM) neuron, a novel mechanism that leverages the inherent hidden states of neuronal membrane potentials to strengthen the selective gating capabilities of SSM. Unlike traditional gating with feed-forward layers (Hochreiter & Schmidhuber, 1997; Chung et al., 2014), temporal chunked LIM neurons naturally accumulate the membrane potential over time, providing a sophisticated yet computationally efficient solution for temporal processing without requiring additional learnable parameters or complex recurrent structures. Furthermore, we design the LIM neuron with continuous flow of membrane information across layers, where each neuron transfers averaged membrane states to initialize neurons in subsequent layers, creating uninterrupted temporal processing throughout the network. Our approach combines LIM neurons with strategically placed LoRA on input and output projections, creating a comprehensive PEFT method specifically tailored for Mamba models. Consequently, ***Memba*** achieves superior performance to existing works (Yoshimura et al., 2024; Halloran et al., 2024) on Mamba fine-tuning while requiring only a fraction of the trainable parameters.

To demonstrate the effectiveness of membrane-driven gating in SSM architectures, Figure 1 presents an overall comparison between a standard SSM model and the proposed ***Memba***. In Figure 1(a), ***Memba*** exhibits sharper saliency focused along the true path, while the original SSM's attention remains diffused, highlighting the inherent benefit of membrane dynamics in promoting selective information flow. Figure1(b) shows the fine-tuning performance comparisons across language and vision tasks, where ***Memba*** consistently outperforms existing PEFT methods. The main contributions of our work are as follows:

- We propose ***Memba***, a membrane-driven PEFT approach that enhances Mamba's gating mechanisms, effectively introducing temporal adaptation without modifying the core state-space components.

- We introduce a temporal chunked LIM neuron with cross-layer membrane propagation. This LIM neuron efficiently processes long sequences while preserving temporal information through evolving membrane potentials.

- Through extensive experiments across language (commonsense reasoning) and vision (vision adaptation) tasks, we demonstrate that ***Memba*** consistently achieves *state-of-the-art* performance compared to existing PEFT methods.

## 2 RELATED WORKS

### 2.1 STATE SPACE MODEL

State Space Models (SSMs) present a promising direction in sequence modeling that combines the parallel processing advantages of Transformer (Vaswani et al., 2017) with the recurrence properties of Recurrent Neural Networks (RNNs) (Hochreiter & Schmidhuber, 1997; Chung et al., 2014). (Gu et al., 2021a) introduces structured state space sequence models (S4), which leverages efficient parameterization of continuous-time state space models for sequence modeling, enabling parallel computation. Subsequent architectures, including the Diagonal State Space (DSS) (Gupta et al., 2022), the Gated State Space (GSS) (Mehta et al., 2022), and Hungry Hungry Hippos (H3) (Fu et al., 2022), enhance the expressivity for better performance while simplifying the implementation complexity. Most recently, Mamba (Gu & Dao, 2023; Dao & Gu, 2024) introduces selective SSMs with data-dependent parameters, enabling dynamic adaptation to input context. This innovation has led to competitive performance across several domains (Pióro et al., 2024; Li et al., 2024; Wang et al., 2025; Quan & Li, 2024; Ota, 2024), establishing SSMs as viable alternatives to Transformer architectures for sequence processing. Given its promising performance and efficiency, Mamba shows significant potential to serve as a backbone for pre-trained foundation models (Gu & Dao, 2023; Ham et al., 2024; Hatamizadeh & Kautz, 2024), making efficient adaptation techniques increasingly important for future downstream applications.

### 2.2 PARAMETER-EFFICIENT FINE-TUNING

PEFT methods address the computational and storage challenges of adapting large pre-trained models to downstream tasks. These approaches modify only a small subset of model parameters while keeping the majority frozen. We differentiate the PEFT algorithms into four categories (Han et al., 2024): additive, selective, parameterized, and hybrid fine-tuning. Additive fine-tuning (Houlsby et al., 2019; He et al., 2021; Pfeiffer et al., 2020; Mahabadi et al., 2021) introduce new trainable components while preserving the original weights, including adapters that insert modules between layers, and prompt-based methods (Li & Liang, 2021; Liu et al., 2021; Lester et al., 2021; Liu et al., 2024b) that augment inputs with learnable tokens. Selective fine-tuning identifies and updates only a critical subset of the original parameters, either through unstructured approaches based on importance metrics (Guo et al., 2020; Sung et al., 2021; Xu et al., 2021) or structured methods targeting specific components (Zaken et al., 2021; He et al., 2023). Reparameterized fine-tuning transforms the optimization space, primarily through low-rank techniques like LoRA (Hu et al., 2022; Zhang et al., 2023) that decompose weight updates into smaller matrices, and its derivatives that incorporate quantization (Dettmers et al., 2023) or direction-magnitude decoupling (Liu et al., 2024a). Hybrid fine-tuning combines multiple paradigms to leverage their complementary strengths, integrating various strategies to achieve optimal performance-efficiency trade-offs (He et al., 2021; Zhang et al., 2024; Hu et al., 2023).

While PEFT methods have been extensively studied in Transformer architectures, their application to SSM models like Mamba remains relatively unexplored. (Halloran et al., 2024) investigates the application of LoRA to Mamba's state space components, while (Yoshimura et al., 2024) comprehensively evaluates various PEFT techniques for SSMs, revealing unique challenges posed by their selective scanning mechanisms. (Ham et al., 2024) demonstrates that targeting projectors rather than state space components yields superior performance through their diagonal-based adaptation approach, focusing primarily on transfer learning. These initial studies indicate that although conventional PEFT methods can be applied to Mamba with reasonable success, there remains significant room for approaches that better leverage the architectural uniqueness of SSMs. The distinctive temporal processing characteristics of SSM architectures create opportunities for specialized adaptation techniques that enhance information flow control capabilities without modifying the core state-space components. Our work addresses this opportunity by introducing a bio-inspired mechanism designed to incorporate temporal dynamics into the gating mechanism.

## 3 PRELIMINARIES

Mamba (Gu & Dao, 2023) architecture addresses the quadratic computational complexity of transformers by employing SSM with selective mechanisms. The standard continuous-time linear time-invariant

SSM is defined by:

$$\mathbf{h}'(t) = \mathbf{A}\mathbf{h}(t) + \mathbf{B}x(t), \quad y(t) = \mathbf{C}\mathbf{h}(t) + Dx(t), \tag{1}$$

where $x(t) \in \mathbb{R}$ is the input sequence, $\mathbf{h}(t) \in \mathbb{R}^N$ is the hidden state with $N$ being the state dimension that controls the model's representational capacity, and $y(t) \in \mathbb{R}$ is the output. The parameters $\mathbf{A} \in \mathbb{R}^{(N \times N)}, \mathbf{B} \in \mathbb{R}^{(N \times 1)}, \mathbf{C} \in \mathbb{R}^{(1 \times N)}$, and $D \in \mathbb{R}$ define the dynamics of the system. For practical implementation with discrete inputs like tokens, Mamba employs Zero-Order Hold (ZOH) discretization to derive equivalent discrete parameters with step size parameter $\boldsymbol{\Delta}$:

$$\bar{\mathbf{A}} = \exp(\boldsymbol{\Delta}\mathbf{A}), \quad \bar{\mathbf{B}} = (\boldsymbol{\Delta}\mathbf{A})^{-1}(\exp(\boldsymbol{\Delta}\mathbf{A}) - \mathbf{I})\boldsymbol{\Delta}\mathbf{B}, \quad \bar{\mathbf{C}} = \mathbf{C}, \quad \bar{D} = D. \tag{2}$$

This discretization transforms the continuous system into its discrete counterpart called as selective scan (`Scan`):

$$\mathbf{h}_t = \bar{\mathbf{A}}\mathbf{h}_{t-1} + \bar{\mathbf{B}}x_t, \quad y_t = \bar{\mathbf{C}}\mathbf{h}_t + \bar{D}. \tag{3}$$

The key innovation in Mamba lies in its selective parameterization mechanism, where $\boldsymbol{\Delta}$, $\mathbf{B}$, and $\mathbf{C}$ become input-dependent, consequently making $\bar{\mathbf{A}}$, $\bar{\mathbf{B}}$, $\bar{\mathbf{C}}$, and $\bar{D}$ input-dependent as well. This is achieved through a structured parameter matrix that computes these values dynamically based on input context, enabling adaptive behavior while maintaining linear computational complexity with respect to sequence length. Critically, Mamba employs a multiplicative gating mechanism that combines the output of selective scan with a transformed input:

$$\hat{y}_t = y_t \odot \sigma(x_t), \tag{4}$$

where $\sigma(\cdot)$ is a SiLU activation and $\odot$ represents element-wise multiplication. This gating mechanism plays a crucial role in controlling information flow through the network. However, while the selective scan in the SSM branch effectively handles temporal processing with evolving hidden states, (Yoshimura et al., 2024; Ham et al., 2024) have shown that fine-tuning this component directly leads to suboptimal results. We address this limitation by introducing temporal adaptation capability to the gating branch during fine-tuning. Detailed notations for the original Mamba architecture are provided in Appendix A.

## 4 METHODOLOGY

In this section, we propose *Memba*, a membrane-driven PEFT approach for Mamba models that introduces temporal processing in the gating branch during fine-tuning. We first present the overall architecture of the *Memba* block, then introduce its three core components.

### 4.1 OVERALL ARCHITECTURE

We develop *Memba* by integrating three main components: ① *Leaky Integrate Membrane (LIM)*, which provides bio-inspired temporal processing; ② *optimal placement of Low-Rank Adaptations*, which strategically modifies key projection layers; and ③ *cross-layer membrane potential transfer*, which maintains temporal coherence across network depth. These modifications build upon the original Mamba architecture as illustrated in Figure 2.

When processing an input tensor $\mathbf{X}_{\text{input}} \in \mathbb{R}^{B \times L \times D}$, with $B$, $L$, and $D$ representing batch size, length of input sequence, and feature dimension respectively, *Memba* first applies normalization and projection, then divides the resulting tensor into two parallel pathways:

$$\mathbf{X}_{\text{SSM}}, \mathbf{X}_{\text{gate}} = \texttt{Split}(\mathbf{W}_{\text{in}}(\texttt{RMS}(\mathbf{X}_{\text{input}}))), \tag{5}$$

where $\mathbf{W}_{\text{in}}$ is the linear layer for input projection ("in_proj") doubling the channel dimension, `RMS` represents RMS normalization, and the `Split` represents channel-wise division by two. The $\mathbf{X}_{\text{SSM}} \in \mathbb{R}^{B \times L \times D}$ processes information through the selective scan, while the $\mathbf{X}_{\text{gate}} \in \mathbb{R}^{B \times L \times D}$ leverages our LIM mechanism. We detail the LIM mechanism in Section 4.2.

$$\mathbf{Y}_{\text{SSM}} = \texttt{Scan}(\mathbf{X}_{\text{SSM}}), \quad \mathbf{Y}_{\text{gate}} = \sigma(\mathbf{W}_{\text{out}}^{\text{gate}}(\text{LIM}(\mathbf{W}_{\text{in}}^{\text{gate}}(\mathbf{X}_{\text{gate}})))). \tag{6}$$

Here, `Scan` represents the selective scan computation as shown in equation 3, $\sigma$ indicates SiLU activation, and $\mathbf{W}_{\text{in}}^{\text{gate}}$, $\mathbf{W}_{\text{out}}^{\text{gate}}$ are the linear projections before and after the LIM neuron, respectively.

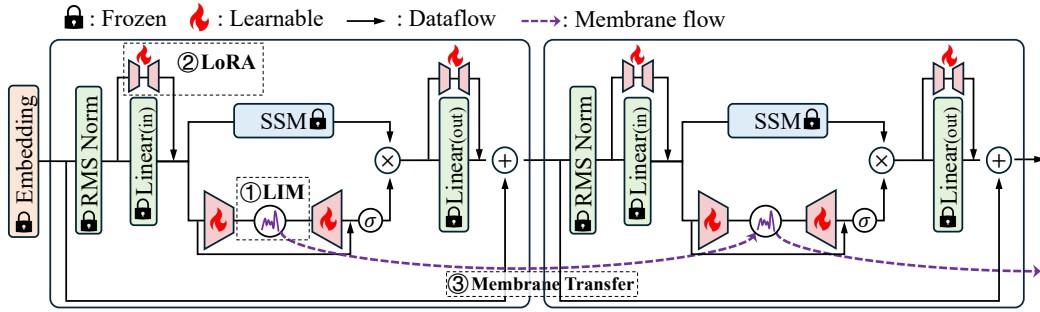

Figure 2: Overview of **Memba** architecture. On top of original Mamba architecture including embedding, normalization, linear layers, and SSM, our **Memba** is designed with ① Leaky Integrate Membrane (LIM), ② Low-Rank Adaptations (LoRAs) on input and output projection, and ③ membrane transfer across layers.

$\mathbf{W}_{\text{in}}^{\text{gate}}$ and $\mathbf{W}_{\text{out}}^{\text{gate}}$ reduce the computational overhead of LIM neuron operations through dimensionality reduction. $\mathbf{Y}_{\text{SSM}}$ and $\mathbf{Y}_{\text{gate}}$ are combined through multiplicative gating to produce the final output:

$$\mathbf{Y}_{\text{out}} = \mathbf{W}_{\text{out}}(\mathbf{Y}_{\text{SSM}} \odot \mathbf{Y}_{\text{gate}}), \tag{7}$$

where $\mathbf{W}_{\text{out}}$ is the linear layer for output projection ("out_proj"). This architecture modification enhances temporal adaptation through the gate path without directly altering the dynamics of the selective scan computation.

## 4.2 KEY COMPONENTS OF *Memba*

### ① LEAKY INTEGRATE MEMBRANE NEURON

**Implementation** To bring the temporal flow to Mamba's gate branch, we introduce the Leaky Integrate Membrane (LIM) neuron shown in Figure 3. The LIM neuron is inspired by the Leaky Integrate-and-Fire (LIF) neuron, described in detail in Appendix B. Rather than processing each token as an individual step, which would be expensive for long sequences, we adopt a chunking strategy for practical implementation. Given an input sequence of length $L$, we partition it into $T$ equal-sized chunks $\{X[1], X[2], ..., X[T]\}$, where each chunk $X[i] \in \mathbb{R}^{B \times L[i] \times D}$ for $i \in \{1, 2, ..., T\}$ contains $L[i] = \lfloor L/T \rfloor$ tokens, with $B$ and $D$ representing batch size and feature dimension respectively. Here, $\lfloor \cdot \rfloor$ denotes the floor function, ensuring each chunk contains the same number of tokens for consistent processing. For this uniformity in chunk size, we trim any remainder tokens when the sequence length is not evenly divisible by the number of chunks.

The key innovation in our LIM approach lies in maintaining membrane continuity across chunk boundaries while processing chunks sequentially. For each chunk (corresponding to one step), we process the input tokens using the leaky integrate membrane dynamics with reset:

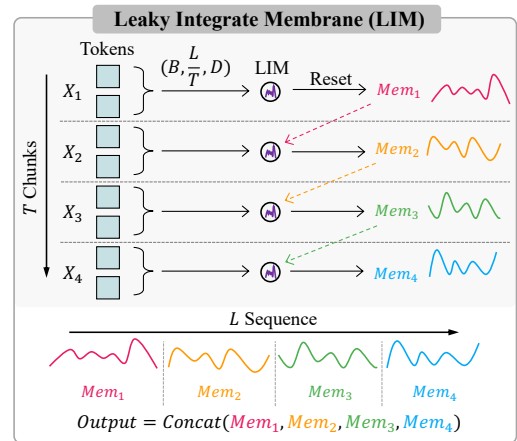

Figure 3: Overview of Leaky Integrate Membrane (LIM). Each token chunk is processed with LIM dynamics, and membrane outputs are concatenated to form the final sequence representation. In this figure, the input contains $L = 8$ tokens split into $T = 4$ chunks, with each chunk ($X_1, X_2, X_3, X_4$) containing 2 tokens.

$$\mathbf{u}[i+1]^l = r(\tau \mathbf{u}[i]^l + \mathbf{W}^l X[i]), \tag{8}$$

$$r(x) = \begin{cases} 0 & \text{if } x > V_{th}, \\ x & \text{otherwise} \end{cases}, \tag{9}$$

Figure 4: Membrane-driven temporal processing in **Memba**. (a) The input image is divided into spatial chunks and flattened into a sequential representation. (b) Membrane distribution and (c) saliency map through the LIM neuron show how the LIM neuron tracks main features while progressively decreasing baseline potentials across chunks, demonstrating adaptive temporal attention. We provide more visualizations in Appendix I.

where $\mathbf{u}[i]^l \in \mathbb{R}^{B \times \lfloor L/T \rfloor \times D}$ is the membrane potential in $l$-th layer at $i$-th chunk, $\tau \in (0, 1]$ is the leaky factor for membrane potential leakage, $\mathbf{W}^l$ is the weight of $l$-th layer, and $r(\cdot)$ is the reset function which sets values above threshold to 0 while preserving others. Note that $\tau$ and $V_{th}$ are key parameters for deciding the membrane distribution, and we analyze the effects of $\tau$ and $V_{th}$ in Section 5.3. After processing all chunks sequentially, we concatenate the outputs to reconstruct the full sequence representation of length $L$, zero padding if necessary to restore any trimmed tokens. The detailed algorithm of LIM neuron is shown in Appendix C.

**Membrane-Driven Temporal Processing** The LIM neuron is designed to retain the temporal information with selective memory through membrane dynamics. We analyze the membrane potential behavior of the LIM neuron as shown in Figure 4. The input is divided into four colored chunks, flattened into a sequence, and processed through the LIM neuron to generate membrane potentials. This visualization reveals two key characteristics of our approach. (1) Critical path features, highlighted in purple and pink boxes, generate pronounced peaks in the membrane potential, demonstrating the model's selective attention to task-relevant information. (2) We observe a gradual decrease in baseline membrane potential across chunks, indicating progressive forgetting of memory as context accumulates. This downward trend aligns with SSM's natural behavior of retaining recent tokens while forgetting earlier ones. Unlike the linear gate in the original Mamba, which delivers uniform sensitivity, our membrane-based approach in **Memba** naturally modulates temporal responsiveness.

The LIM neuron not only enables efficient processing of long sequences but also provides theoretical advantages for model optimization and generalization as follows:

**Theorem 1.** *Let $\mathcal{L}(\mathbf{y})$ be a twice-differentiable loss function, and let $\mathbf{y}_t = f_\theta(\mathbf{X}_t)$ be the output of a standard Mamba block. When augmented with our LIM mechanism, the effective output becomes $\hat{\mathbf{y}}_t = \mathbf{y}_t \odot g(\mathbf{u}_t)$, where $\mathbf{u}_t$ is the membrane potential. The expected loss satisfies:*

$$\mathbb{E}[\mathcal{L}(\hat{\mathbf{y}}_t)] = \mathcal{L}(\mathbf{y}_t \odot g(\bar{\mathbf{u}}_t)) + \mathcal{R}(\mathbf{y}_t, \bar{\mathbf{u}}_t) + \mathcal{O}(\|\varepsilon_t\|^3) \tag{10}$$

*where $\bar{\mathbf{u}}_t = \mathbb{E}[\mathbf{u}_t]$, $\varepsilon_t = \mathbf{u}_t - \bar{\mathbf{u}}_t$ with $\mathbb{E}[\varepsilon_t] = 0$, and $\mathcal{R}$ is a bounded regularization term satisfying $\mathcal{R}(\mathbf{y}_t, \bar{\mathbf{u}}_t) \leq \frac{\gamma}{2} \cdot \lambda_{\max} \cdot \epsilon^2$, where $\lambda_{\max}$ is the maximum eigenvalue of the Hessian of $\mathcal{L}$ and $\gamma$ depends on the model outputs and gate sensitivity.*

This theorem reveals LIM's dual effect: the mean membrane component provides temporal context integration through leaky dynamics, while the fluctuation component introduces bounded regularization that adapts to model sensitivity and output magnitude. The smoother loss landscape geometry observed in Appendix D.6 represents empirical evidence of how this theoretical regularization manifests in practice. See Appendix D for complete derivation.

②  PLACEMENT OF LOW-RANK ADAPTATIONS

To identify the optimal application points for our LIM neuron, we conduct an ablation study on the main components of **Memba**-130M. We evaluate applying LoRA to different projectors on commonsense reasoning tasks. "All" refers to simultaneously adapting four key projectors in original Mamba: input ("in_proj"), output ("out_proj"), time-scale ("dt_proj"), and selective state ("x_proj") as shown in Appendix A. We then systematically exclude individual components, denoted by "−dt", "−x", "−out", and "−in". For example, in "−dt", LoRAs are applied to in_proj, out_proj, and x_proj.

The results in Table 1 clearly show that input and output projectors are most critical for fine-tuning with the LIM neuron. Excluding either leads to performance drops of 1.2% and 0.8% respectively, suggesting these projectors act as crucial information bottlenecks in the *Memba* architecture. We further compare the accuracy of full fine-tuning against *Memba* with LoRA applied to both `in_proj` and `out_proj` on 790M and 1.4B architectures in Figure 5. Notably, our approach achieves higher performance than full fine-tuning while using fewer trainable parameters. Full fine-tuning often suffers from overfitting due to the relatively large number of trainable parameters compared to the size of downstream task datasets (Ham et al., 2024).

Table 1: Ablation study on the impact of applying LoRA to different projection components in *Memba*-130M.

|  | All | −dt | −x | −out | −in |
|---|---|---|---|---|---|
| Avg. Acc. (%) | 43.9 | 43.9 | 43.7 | 43.1 | 42.7 |
| Acc. drop (%) | - | 0.0 | 0.2 ↓ | **0.8** ↓ | **1.2** ↓ |

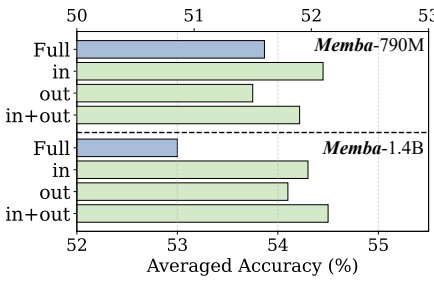

Figure 5: Performance comparison between full fine-tuning and *Memba* with Lo-RAs applied to `in_proj` and `out_proj` across 790M and 1.4B models.

### ③ CROSS-LAYER MEMBRANE TRANSFER

As large models scale in depth, maintaining temporal context across layers becomes challenging, yet crucial for effective sequence modeling. To address this, we implement a cross-layer membrane potential transfer technique that propagates temporal information throughout the network hierarchy without increasing computational cost. After processing all chunks within a layer and obtaining membrane potentials $\{\mathbf{u}^l[1], \mathbf{u}^l[2], ..., \mathbf{u}^l[T]\}$, we compute the average membrane state, $\bar{\mathbf{u}}^l$ across all chunks. This averaged membrane potential $\bar{\mathbf{u}}^l$ serves as the initial state for the first chunk of the subsequent layer:

$$\bar{\mathbf{u}}^l = \frac{1}{T}\sum_{i=1}^{T}\mathbf{u}^l[i], \quad \mathbf{u}^{l+1}[1] = \bar{\mathbf{u}}^l. \tag{11}$$

By transferring this compressed temporal context, we enable each layer to begin processing with a summary of the temporal dynamics captured by the previous layer. This mechanism creates a hierarchical flow of temporal information through the network, allowing deeper layers to build upon the representations learned by earlier layers. The use of averaged membrane potentials helps prevent information loss that might occur if only the final state were propagated, ensuring that the model maintains sensitivity to patterns that emerged at different points in the sequence.

## 5 EXPERIMENTS

In this section, we evaluate the proposed *Memba* approach on both language and vision tasks. For language tasks, we fine-tune pre-trained Mamba on 8 commonsense reasoning benchmarks. For vision tasks, we fine-tune two different architectures, Vim (Zhu et al., 2024) and VMamba (Liu et al., 2024c), on the Visual Task Adaptation-Benchmark (VTAB)-1k dataset (Zhai et al., 2019). We also present ablation studies to analyze the *Memba* architecture for a better understanding. Our experimental setup follows the framework established in MambaPeft (Yoshimura et al., 2024), and the experiments are implemented by A100 GPUs.

### 5.1 LANGUAGE TASK

**Experimental Details** To evaluate *Memba*, we begin by loading pre-trained Mamba models (Gu & Dao, 2023) trained on the Pile dataset (Gao et al., 2020), apply the *Memba* architecture modifications, and then fine-tune *Memba* on a combined dataset of approximately 170k examples from commonsense reasoning tasks. We evaluate performance across eight individual benchmarks: BoolQ, PIQA, SIQA, HellaSwag, WinoGrande, ARC-Challenge, ARC-Easy, and OpenbookQA. For fine-tuning, we follow the setup of LLM-Adapter (Hu et al., 2023), and for evaluation, we use the `lm_eval` framework (Gao et al., 2024). Detailed hyperparameters are provided in Appendix E.

Table 2: Performance comparison between *Memba* and prior methods on commonsense reasoning datasets. Reported values are accuracy percentages (%). **Bold** and underlined entries indicate the best and second-best performance, respectively. HS and WG refer to the HellaSwag and WinoGrande datasets. All results are from (Yoshimura et al., 2024), except for SLL LoRA (Halloran et al., 2024).

| Model | Method | #Params(%) | BoolQ | PIQA | SIQA | HS | WG | ARC-e | ARC-c | OBQA | Avg. |
|---|---|---|---|---|---|---|---|---|---|---|---|
| Pythia 160M | Full | 100 | 61.3 | 62.9 | 37.1 | 30.7 | 50.6 | 41.5 | 24.3 | 27.8 | 42.0 |
| | LoRA | 0.72 | 61.0 | 62.0 | 36.3 | 30.3 | 52.0 | 38.2 | 24.6 | 28.0 | 41.6 |
| Mamba 130M | Full | 100 | 56.1 | 65.3 | 38.7 | 35.3 | 52.0 | 46.4 | 25.7 | 32.8 | 43.8 |
| | SLL LoRA | 1.45 | 56.3 | 63.3 | 38.2 | 34.6 | 51.6 | 43.5 | 23.6 | 30.6 | 42.7 |
| | Additional-scan | 0.51 | 57.8 | 64.1 | 37.5 | 34.5 | 53.0 | 41.3 | 23.5 | 30.0 | 42.7 |
| | Affix-tuning | 64.64 | 59.7 | 64.3 | 38.2 | **35.2** | 51.9 | 42.9 | 24.0 | 29.0 | 43.2 |
| | LoRA (in_proj) | 2.23 | 53.5 | 62.9 | 38.2 | 33.8 | **53.1** | 46.4 | 23.7 | 30.8 | 42.8 |
| | LoRA$_p$ (X) | 2.67 | **61.7** | 64.0 | 39.5 | 34.3 | 52.2 | 43.5 | **25.3** | 29.4 | 43.7 |
| | *Memba* (in_proj) | 3.95 | 56.3 | 64.4 | 37.7 | 34.3 | 52.4 | 48.9 | 23.8 | 30.0 | 43.5 |
| | *Memba* (out_proj) | 3.10 | 58.4 | 64.9 | 38.8 | 34.4 | 51.8 | 50.0 | 24.2 | 30.0 | 44.0 |
| | *Memba* (in+out_proj) | 5.20 | 58.8 | **65.8** | 40.1 | 34.7 | 51.6 | 47.7 | 24.7 | **31.2** | **44.3** |
| Pythia 410M | Full | 100 | 55.0 | 68.4 | 42.1 | 40.8 | 53.9 | 50.8 | 26.7 | 30.0 | 46.0 |
| | LoRA | 0.77 | 61.3 | 67.7 | 40.8 | 39.2 | 54.9 | 48.1 | 24.7 | 28.6 | 45.7 |
| Mamba 370M | Full | 100 | 58.1 | 69.9 | 41.9 | 45.7 | 53.8 | 52.7 | 29.7 | 33.4 | 48.2 |
| | SLL LoRA | 2.30 | 59.5 | 69.6 | 42.2 | 44.1 | 54.9 | 50.6 | 26.3 | 30.8 | 47.3 |
| | Additional-scan | 0.47 | 61.9 | 69.3 | 41.2 | 45.3 | 54.9 | 49.5 | 28.4 | 31.4 | 47.7 |
| | Affix-tuning | 68.88 | 61.2 | 68.4 | 39.6 | 46.2 | 55.4 | 48.2 | 28.2 | 30.6 | 47.2 |
| | LoRA (in_proj) | 2.07 | 55.4 | 68.6 | 41.0 | 44.7 | 54.1 | 52.4 | 28.3 | 33.4 | 47.2 |
| | LoRA$_p$ (X) | 2.67 | 60.8 | 68.8 | 42.1 | 44.7 | **56.2** | 50.4 | 27.4 | 32.2 | 47.8 |
| | *Memba* (in_proj) | 3.67 | 59.1 | 69.2 | **42.7** | 45.1 | 54.0 | 55.2 | 28.0 | 33.0 | 48.3 |
| | *Memba* (out_proj) | 2.88 | 58.0 | **70.0** | 42.5 | 45.4 | 54.9 | 55.4 | 26.7 | 31.2 | 48.0 |
| | *Memba* (in+out_proj) | 4.83 | 58.7 | 69.8 | 42.5 | 45.4 | 53.7 | 55.6 | 28.0 | **34.0** | **48.5** |
| Pythia 1B | Full | 100 | 55.0 | 70.2 | 42.5 | 47.5 | 54.4 | 54.1 | 29.7 | 33.2 | 48.3 |
| | LoRA | 0.41 | 60.0 | 69.3 | 40.9 | 45.3 | 53.6 | 49.8 | 27.2 | 31.0 | 47.1 |
| Mamba 790M | Full | 100 | 62.0 | 72.1 | 44.8 | 54.0 | 55.9 | 57.7 | 31.2 | 35.2 | 51.6 |
| | SLL LoRA | 3.1 | 60.7 | 72.0 | 42.4 | 54.7 | 56.9 | 55.3 | 29.4 | 34.2 | 50.7 |
| | Additional-scan | 0.33 | **63.0** | 71.9 | 41.9 | 54.2 | 57.1 | 54.9 | 30.0 | 32.6 | 50.7 |
| | Affix-tuning | 69.99 | 61.0 | 72.5 | 41.0 | 54.9 | 55.6 | 54.6 | 29.6 | 33.8 | 50.4 |
| | LoRA (in_proj) | 1.47 | 61.7 | 71.9 | 44.0 | 50.8 | 56.7 | 56.3 | 30.5 | 33.8 | 50.7 |
| | LoRA$_p$ (X) | 1.75 | 59.9 | 72.2 | 44.2 | 52.8 | **58.0** | 53.7 | 30.8 | **34.8** | 50.8 |
| | *Memba* (in_proj) | 2.61 | 62.2 | 72.6 | 43.8 | 54.4 | 57.5 | 61.0 | **31.7** | 34.0 | 52.1 |
| | *Memba* (out_proj) | 2.04 | 57.9 | 72.0 | **44.3** | 55.2 | 56.7 | 60.4 | 31.2 | 34.0 | 51.5 |
| | *Memba* (in+out_proj) | 3.45 | 62.4 | **72.8** | 44.1 | 54.8 | 57.3 | 61.3 | 31.6 | 34.3 | **52.3** |
| Pythia 1.4B | Full | 100 | 58.6 | 71.1 | 42.7 | 53.6 | 55.1 | 58.5 | 29.9 | 34.8 | 50.5 |
| | LoRA | 0.44 | 60.1 | 71.3 | 42.5 | 50.1 | 58.9 | 57.6 | 29.6 | 33.6 | 50.5 |
| Mamba 1.4B | Full | 100 | 61.4 | 73.3 | 43.9 | 56.9 | 59.0 | 59.7 | 34.0 | 35.4 | 53.0 |
| | SLL LoRA | 4.64 | 59.7 | 73.5 | 43.1 | 56.9 | 60.7 | 59.7 | 31.7 | 36.0 | 52.7 |
| | Additional-scan | 0.26 | 63.0 | 73.5 | 42.8 | 57.5 | 60.5 | 60.9 | 32.4 | **37.4** | 53.5 |
| | LoRA (in_proj) | 1.13 | 62.6 | 73.6 | **43.7** | 55.6 | 59.7 | 58.3 | 31.7 | 35.6 | 52.6 |
| | LoRA$_p$ (X) | 1.36 | 63.1 | 73.5 | 42.7 | 57.7 | 61.6 | 60.4 | 32.9 | **37.4** | 53.7 |
| | *Memba* (in_proj) | 2.02 | 63.1 | 74.0 | 43.0 | 58.5 | **61.7** | 63.8 | 33.2 | **37.4** | 54.3 |
| | *Memba* (out_proj) | 1.58 | 62.6 | **74.3** | 43.5 | 58.6 | 60.6 | 64.3 | 32.3 | 37.0 | 54.1 |
| | *Memba* (in+out_proj) | 2.68 | **64.4** | **74.3** | 43.4 | 58.6 | 60.7 | 64.2 | 33.0 | **37.4** | **54.5** |

**Results** The overall results of *Memba* on language tasks are shown in Table 2. Our comparison baselines are Transformer-based architecture Pythia (Biderman et al., 2023) and previous Mamba fine-tuning approaches, including SLL LoRA (Halloran et al., 2024) and MambaPEFT (Yoshimura et al., 2024). We present three variants of our approach for each model size: LoRA applied to input projections (in_proj), output projections (out_proj), and both. We observe that our *Memba* consistently achieves *state-of-the-art* performance. Notably, with the Mamba 790M model, our in_proj+out_proj variant demonstrates a substantial 1.5% absolute improvement over the best results reported in MambaPEFT (Yoshimura et al., 2024), highlighting the effectiveness of our membrane-based approach for larger models. The iso-parameter cases are shown in the Appendix G.2.

## 5.2 VISION TASK

**Experimental Details** We further investigate the effectiveness of *Memba* on vision tasks by fine-tuning two pre-trained Mamba architectures: Vim (Zhu et al., 2024) and VMamba (Hatamizadeh & Kautz, 2024), both initially trained on ImageNet-1k (Deng et al., 2009). Since the original VMamba architecture removes the gate branch, we utilize Vanilla-VMamba, which preserves the gating

Table 3: Performance comparison between *Memba* and previous works on VTAB-1k datasets. Values shown are accuracy percentages (%). **Bold** and underlined values represent the best and second-best performance respectively. † represents our implementation, and other results are from (Yoshimura et al., 2024).

| Model | Method | #Params(K) | Natural | Specialized | Structured | Avg. |
|---|---|---|---|---|---|---|
| ViT-S | Scratch | 21,704 | 10.66 | 56.12 | 24.83 | 26.20 |
| | Full | 21,704 | 51.79 | 72.29 | 45.27 | 53.47 |
| | LoRA | 628 | 73.60 | 82.22 | 57.61 | 68.68 |
| | Adaptformer | 333 | 73.63 | 83.15 | 57.80 | 68.97 |
| | Adapter+ | 122 | 74.68 | 83.57 | 58.82 | 69.87 |
| Vim-S | Scratch | 25,450 | 8.33 | 49.87 | 28.16 | 25.42 |
| | Full | 25,450 | 59.35 | 68.74 | 34.39 | 50.08 |
| | LoRA (embed) | 45 | 64.66 | 77.53 | 43.83 | 58.60 |
| | LoRA (x_proj) | 2,540 | 74.41 | 81.92 | 54.88 | 67.77 |
| | LoRA (dt_proj) | 2,442 | 75.35 | 83.05 | 57.12 | 69.30 |
| | LoRA (out_proj) | 2,663 | 76.42 | 83.96 | 60.08 | 71.12 |
| | LoRA (in_proj) | 1,483 | 76.58 | 84.08 | 60.16 | 71.25 |
| | LoRA$_p$ (Z) | 1,778 | 76.15 | 84.26 | 59.72 | 70.94 |
| | LoRA$_p$ (X) | 1,778 | 76.64 | 83.89 | 60.84 | 71.52 |
| | LoRA (in+out_proj) | 709 | 75.69 | 84.42 | 59.43 | 70.68 |
| | Hybrid (w/ proj) | 117,236 | 77.00 | 84.41 | 61.55 | 72.05 |
| | Hybrid (w/o proj) | 1,044 | 76.85 | 84.42 | 61.06 | 71.80 |
| | *Memba* (in_proj) | 2,064 | 76.91 | 85.10 | 60.70 | 71.81 |
| | *Memba* (out_proj) | 3,244 | 76.92 | 85.32 | 61.18 | 72.06 |
| | *Memba* (in+out_proj) | 4,718 | **77.07** | **85.66** | **61.70** | **72.40** |
| Vanilla-VMamba-S | LoRA (in_proj)† | 3,993 | 77.76 | 86.05 | 63.44 | 73.48 |
| | LoRA (out_proj)† | 2,396 | 77.43 | 86.06 | 64.33 | 73.73 |
| | LoRA (in+out_proj)† | 6,389 | 77.31 | 85.81 | 63.30 | 73.20 |
| | *Memba* (in_proj) | 5,591 | 77.66 | 86.06 | 63.92 | 73.64 |
| | *Memba* (out_proj) | 3,993 | **77.77** | 85.98 | **64.87** | **74.07** |
| | *Memba* (in+out_proj) | 7,987 | 77.14 | **86.14** | 63.93 | 73.48 |

mechanism. We evaluate performance on the VTAB-1k image classification benchmark (Zhai et al., 2019), which comprises Natural, Specialized, and Structured domains. For fine-tuning, we adopt the DeiT (Touvron et al., 2021) training framework across all domains. Detailed hyperparameters are provided in Appendix E.

**Results** Table 3 presents our *Memba* results on Vim (Zhu et al., 2024) and VMamba (Hatamizadeh & Kautz, 2024) architectures, compared against ViT (Dosovitskiy et al., 2020) and previous PEFT approaches from MambaPEFT (Yoshimura et al., 2024). As in our language experiments, we evaluate variants with LoRA applied to input projections (in_proj), output projections (out_proj), or both. Our *Memba* outperforms previous PEFT methods on both Vim-S and Vanilla-VMamba-S architectures. Notably, with Vim-S, our out_proj variant achieves 72.40% average accuracy across all domains, surpassing the previous best result of Hybrid method while using only 28% of the trainable parameters. Per-task performances of each categories are shown in Appendix F.

## 5.3 ANALYSIS

**Contribution of Key Components** To quantify the impact of each component in *Memba*, we conduct an ablation study examining combinations of our three key innovations: ① LIM neurons, ② LoRA, and ③ membrane transfer. Table 4(a) presents these results. For language tasks, we use the 130M parameter Mamba model on commonsense reasoning, while for vision tasks, we use the Vim-S architecture on the Specialized category of VTAB-1k, which includes Camelyon, EuroSAT, Resisc45, and Retinopathy datasets. Our results show that while LoRA enables effective fine-tuning of Mamba models, adding LIM neurons and membrane transfer further boosts performance.

**Impact of Membrane Parameters** In LIM neuron, two key hyperparameters control the membrane dynamics: the leaky factor ($\tau$) and threshold ($V_{th}$) in equation 8 and equation 9. The leaky factor determines how much previous membrane potential is retained, while the threshold governs the reset mechanism. To understand their influence, we conduct experiments with various parameter combinations using the Vim-S architecture on the Specialized category of VTAB-1k, as shown in

Table 4: Ablation studies of (a) key components of LIM and (b) membrane parameters.

| ① LIM | ② LoRA | ③ Membrane Transfer | Mamba-130m | Vim-S |
|---|---|---|---|---|
| ✓ | ✗ | ✗ | 43.1 | 83.8 |
| ✓ | ✗ | ✓ | 43.3 | 84.0 |
| ✗ | ✓ | ✗ | 43.8 | 85.3 |
| ✓ | ✓ | ✗ | 44.0 | 85.5 |
| ✓ | ✓ | ✓ | **44.3** | **85.7** |

| $\tau$ ($V_{th} = 1$) | $1/2$ | $1/3$ | $1/4$ | $1/5$ |
|---|---|---|---|---|
| Averaged Acc.(%) | **85.7** | 85.2 | 85.2 | 85.0 |
| $V_{th}$ ($\tau = 1/2$) | 0.5 | 1 | 2 | 3 |
| Averaged Acc.(%) | 85.4 | **85.7** | 85.3 | 85.2 |

(a) Accuracy comparison between combinations of key components of LIM neuron.

(b) Impact of membrane parameters on vision tasks. $\tau$ and $V_{th}$ are leaky factor and threshold respectively.

Table 4(b). For the leaky factor, higher values of $\tau$ yield better performance, confirming that stronger retention of previous states benefits temporal processing. For the threshold, $V_{th} = 1$ provides optimal performance, indicating lower values trigger excessive resets that disrupt information flow, while higher values reduce reset frequency, hindering the model's ability to filter irrelevant information. These results demonstrate that balanced membrane dynamics are critical for effective temporal processing. Additional extensive ablation studies are presented in Appendix G.

## 6 LIMITATIONS

The LIM neuron accumulates membrane dynamics with a reset function, which inherently introduces recurrent computation and consequently incurs additional computational overhead. Although the selective scan operation in SSMs also involves recurrent computation, Gu et al. (Gu & Dao, 2023) address this efficiency challenge through specialized hardware kernel modifications. Similarly, our LIM algorithm can be integrated into optimized CUDA kernels in future implementations. The widely-used SpikingJelly framework (Fang et al., 2023) demonstrates that custom CUDA kernels for LIF neurons achieve up to $30\times$ speedups through operator fusion, where element-wise operations (leaky decay, addition, reset) are fused into a single kernel launch. Our chunk-based recurrent operations can directly leverage this optimization approach, as the sequential dependency between chunks does not prevent intra-chunk parallelization. We expect this would achieve negligible computational overhead comparable to the selective scan kernel.

## 7 CONCLUSION

We introduce *Memba*, a membrane-driven PEFT approach for Mamba models that integrates Leaky Integrate Membrane (LIM) neurons with strategic Low-Rank Adaptations (LoRAs). Without modifying state-space components, our method enhances temporal adaptation capabilities exclusively through the gating branch, preserving the tuned dynamics of pre-trained SSMs. Experiments across language and vision tasks demonstrate *Memba*'s consistent improvement over existing PEFT methods. Our approach represents an important step toward specialized adaptation techniques for SSMs, opening possibilities for effective fine-tuning of these architectures across diverse applications.

## ACKNOWLEDGEMENT

This work was supported in part by CoCoSys, a JUMP2.0 center sponsored by DARPA and SRC, the National Science Foundation (CAREER Award, Grant #2312366, Grant #2318152), the DARPA Young Faculty Award, the DoE MMICC center SEA-CROGS (Award #DE-SC0023198) and the Global Industrial Technology Cooperation Center (GITCC) program.

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

## A    ORIGINAL MAMBA ARCHITECTURE

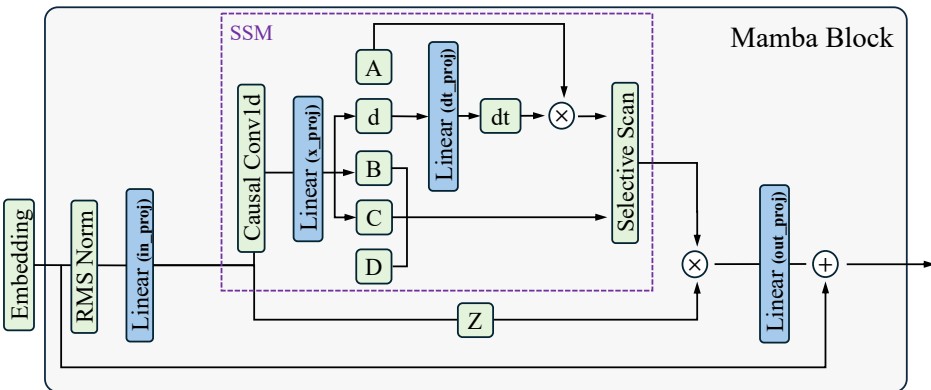

Figure 6: Overall architecture of the original Mamba block.

To facilitate understanding of our LoRA placement strategy in conjunction with the LIM neuron, we present the overall architecture of the original Mamba block in Figure 6. The SSM core, enclosed in the purple dotted box, follows the computations described in Equations equation 1, equation 2, equation 3, where $Z$ represents the gating values. We highlight the four linear layers used for LoRA insertion in blue: in_proj, x_proj, dt_proj, and out_proj. The ablation results reported in Table 1 correspond to the impact of selectively applying LoRA to each of these layers in combination with the LIM neuron.

## B    LEAKY INTEGRATE-AND-FIRE NEURON

The Leaky Integrate-and-Fire (LIF) neuron (Burkitt, 2006) has emerged as an important component for energy-efficient computation in SNNs (Maass, 1997; Roy et al., 2019). The LIF neuron processes temporal information through its membrane potential dynamics as follows:

$$\mathbf{u}[i+1]^l = \tau\mathbf{u}[i]^l + \mathbf{W}^l r(\mathbf{u}[i]^{l-1}), \tag{12}$$

$$r(\mathbf{u}[i]^l) = \begin{cases} 1 & \text{if } \mathbf{u}[i]^l > V_{th}, \\ 0 & \text{otherwise} \end{cases}, \tag{13}$$

where, $\mathbf{u}[i]^l$ is the membrane potential in $l$-th layer at timestep $i$, $\tau \in (0,1]$ is the leaky factor for membrane potential leakage, $\mathbf{W}^l$ is the weight of $l$-th layer, and $f(\cdot)$ is the LIF function with threshold $V_{th}$. When the membrane $\mathbf{u}[i]^l$ is higher than $V_{th}$, the LIF function generates a spike and the membrane potential is reset to 0. The LIF neuron exhibits two core characteristics: (1) Leaky Integration and (2) Reset Mechanism.

The term $\tau\mathbf{u}[i]^l$ models the leaky accumulation of membrane potential in biological neurons' membranes. This leaky mechanism enables forgetting old, unnecessary information while focusing on new inputs, mimicking how biological neurons naturally attenuate stale signals. The reset function $r(\cdot)$ is analogous to the fire-and-reset mechanism in biological neurons, where the neuron discharges its potential upon crossing a threshold $V_{th}$. In biological neurons, this generates a spike.

## C    THE DETAILS OF LIM ALGORITHM

We present the details of the LIM process, which involves efficient processing of sequential data through temporal chunking and cross-layer membrane potential transfer.

---

**Algorithm 1** Leaky Integrate Membrane

---

**Input:** Input sequence tensor $\mathbf{X}_{\text{input}} \in \mathbb{R}^{B \times L \times D}$, number of chunks $T$, leak factor $\tau$, threshold $V_{\text{th}}$, linear projection $f(\cdot)$, previous membrane potential $\mathbf{u}_{\text{prev}} \in \mathbb{R}^{B \times \lfloor L/T \rfloor \times D}$

**Output:** Output membrane $\mathbf{u}_{\text{output}} \in \mathbb{R}^{B \times L \times D}$, averaged membrane $\mathbf{u}_{\text{prev}} \in \mathbb{R}^{B \times \lfloor L/T \rfloor \times D}$

$l \leftarrow \lfloor L/T \rfloor$            *# Chunk size using floor function*
$r \leftarrow L - l \times T$            *# Calculate remainder*
**if** $r \neq 0$ **then**
    $\mathbf{X} \leftarrow \mathbf{X}_{\text{input}}[:, : -r, :]$        *# Trim sequence to be divisible by T*
    $L \leftarrow L - r$
**end if**
**if** $\mathbf{u}_{\text{prev}}$ is None **then**
    $\mathbf{u}_{\text{current}} \leftarrow \mathbf{0}$
**else**
    $\mathbf{u}_{\text{current}} \leftarrow \mathbf{u}_{\text{prev}}$        *# Transfer membrane state from previous layer*
**end if**
**for** $i = 0$ **to** $T - 1$ **do**
    start_idx $\leftarrow i \times l$
    end_idx $\leftarrow (i + 1) \times l$
    $\mathbf{X}_i \leftarrow \mathbf{X}[:, \text{start\_idx} : \text{end\_idx}, :]$        *# Extract current chunk*
    $\mathbf{u}_{\text{current}} \leftarrow \tau \cdot \mathbf{u}_{\text{current}} + f(\mathbf{X}_i)$        *# Leaky integrate dynamics*
    $\mathbf{u}_{\text{current}} \leftarrow \text{Reset}(\mathbf{u}_{\text{current}})$        *# Apply Reset with threshold $V_{th}$*
    $\mathbf{u}_{\text{list}}[i] = \mathbf{u}_{\text{current}}$
**end for**
$\mathbf{u}_{\text{output}} \leftarrow \text{concatenate}(\mathbf{u}_{\text{list}}, \dim = 1)$        *# Combine all chunks*
$\mathbf{u}_{\text{prev}} \leftarrow \text{mean}(\mathbf{u}_{\text{list}}, \dim = 1)$        *# Average the membrane for transfer to next layer*
**if** $r \neq 0$ **then**
    $\mathbf{u}_{\text{output}} \leftarrow \text{pad}(\mathbf{u}_{\text{output}})$        *# Pad to original length*
**end if**
**return** $\mathbf{u}_{\text{output}}, \mathbf{u}_{\text{prev}}$

---

# D THEORETICAL DERIVATION

## D.1 DECOMPOSITION OF LIM SIGNAL

In our LIM mechanism, the membrane potential evolves according to the dynamics defined in equation 8:

$$\mathbf{u}_t = r(\tau \mathbf{u}_{t-1} + \mathbf{W}\mathbf{X}_t) \tag{14}$$

where $r(\cdot)$ is the reset function from equation 9.

We decompose the membrane potential $\mathbf{u}_t$ into a mean component and a fluctuation component:

$$\mathbf{u}_t = \bar{\mathbf{u}}_t + \varepsilon_t, \quad \text{where } \mathbb{E}[\varepsilon_t] = \mathbf{0} \tag{15}$$

The mean component $\bar{\mathbf{u}}_t = \mathbb{E}[\mathbf{u}_t]$ captures the expected membrane potential, which represents the temporal integration of input history. The fluctuation component $\varepsilon_t = \mathbf{u}_t - \bar{\mathbf{u}}_t$ represents deviations from this expected behavior.

**Boundedness of membrane potential:** The reset function $r(\cdot)$ enforces an upper bound by resetting values exceeding $V_{th}$ to zero, ensuring $\mathbf{u}_t \leq V_{th}$ element-wise. For the lower bound, we note that the membrane potential can take negative values when $\tau \mathbf{u}_{t-1} + \mathbf{W}\mathbf{X}_t < 0$. However, since $\tau \in (0, 1]$ and inputs are bounded ($\|\mathbf{X}_t\| \leq X_{\max}$), the recursive relation $|\mathbf{u}_t| \leq \tau |\mathbf{u}_{t-1}| + \|\mathbf{W}\|X_{\max}$ yields, by geometric series convergence:

$$-M_u \leq \mathbf{u}_t \leq V_{th} \quad \text{for all } t, \quad \text{where } M_u = \frac{\|\mathbf{W}\|X_{\max}}{1 - \tau} \tag{16}$$

Defining $U_{\max} = \max(V_{th}, M_u)$, the fluctuations are bounded:

$$\|\varepsilon_t\| \leq 2U_{\max} \tag{17}$$

The decomposition $\mathbf{u}_t = \bar{\mathbf{u}}_t + \varepsilon_t$ enables us to analyze how membrane dynamics affect the loss function: the mean component $\bar{\mathbf{u}}_t$ provides stable temporal context integration, while the bounded fluctuation component $\varepsilon_t$ introduces controlled variability that, as we show in the following sections, acts as an adaptive regularization mechanism.

## D.2 MULTIPLICATIVE GATING ANALYSIS

In our architecture, the membrane potential influences the output through multiplicative gating:

$$\hat{\mathbf{y}}_t = \mathbf{y}_t \odot g(\mathbf{u}_t) \tag{18}$$

where $\mathbf{y}_t$ is the output from the selective scan, $g(\cdot)$ is the gating function applied element-wise (typically SiLU activation), and $\odot$ denotes element-wise multiplication.

Using the decomposition of $\mathbf{u}_t$ from the previous section:

$$g(\mathbf{u}_t) = g(\bar{\mathbf{u}}_t + \varepsilon_t) \tag{19}$$

We apply an element-wise Taylor expansion of $g$ around $\bar{\mathbf{u}}_t$:

$$g(\mathbf{u}_t) = g(\bar{\mathbf{u}}_t) + g'(\bar{\mathbf{u}}_t) \odot \varepsilon_t + \frac{1}{2} g''(\bar{\mathbf{u}}_t) \odot \varepsilon_t^2 + O(\|\varepsilon_t\|^3) \tag{20}$$

where $g'(\bar{\mathbf{u}}_t)$ and $g''(\bar{\mathbf{u}}_t)$ denote element-wise derivatives, and $\varepsilon_t^2$ represents element-wise squaring.

Therefore, the gated output becomes:

$$\hat{\mathbf{y}}_t = \mathbf{y}_t \odot g(\bar{\mathbf{u}}_t) + \mathbf{y}_t \odot g'(\bar{\mathbf{u}}_t) \odot \varepsilon_t + \frac{1}{2} \mathbf{y}_t \odot g''(\bar{\mathbf{u}}_t) \odot \varepsilon_t^2 + O(\|\varepsilon_t\|^3) \tag{21}$$

## D.3 ANALYSIS OF EXPECTED LOSS

We analyze the expected loss by adopting a local perturbation perspective: for a given input neighborhood, we treat the selective scan output $\mathbf{y}_t$ as approximately constant and examine how variations in the membrane potential $\mathbf{u}_t$ affect the loss. This is justified by the architectural separation between the SSM branch and the gate branch, which process information through distinct temporal dynamics.

We expand around $\mathbf{y}_t^* = \mathbf{y}_t \odot g(\bar{\mathbf{u}}_t)$, which represents the output when the membrane potential is at its local mean value. Note that this is not exactly $\mathbb{E}[\hat{\mathbf{y}}_t]$ due to the nonlinearity of $g$, but serves as a natural reference point that isolates the effects of membrane fluctuations.

Let $\boldsymbol{\delta}\mathbf{y}_t = \hat{\mathbf{y}}_t - \mathbf{y}_t^*$. From the multiplicative gating analysis:

$$\boldsymbol{\delta}\mathbf{y}_t = \mathbf{y}_t \odot g'(\bar{\mathbf{u}}_t) \odot \varepsilon_t + \frac{1}{2} \mathbf{y}_t \odot g''(\bar{\mathbf{u}}_t) \odot \varepsilon_t^2 + O(\|\varepsilon_t\|^3) \tag{22}$$

The Taylor expansion of the loss function around $\mathbf{y}_t^*$ gives:

$$\mathcal{L}(\hat{\mathbf{y}}_t) = \mathcal{L}(\mathbf{y}_t^* + \boldsymbol{\delta}\mathbf{y}_t) \tag{23}$$

$$= \mathcal{L}(\mathbf{y}_t^*) + \nabla\mathcal{L}(\mathbf{y}_t^*)^\top \boldsymbol{\delta}\mathbf{y}_t + \frac{1}{2} \boldsymbol{\delta}\mathbf{y}_t^\top \nabla^2 \mathcal{L}(\mathbf{y}_t^*) \boldsymbol{\delta}\mathbf{y}_t + O(\|\boldsymbol{\delta}\mathbf{y}_t\|^3) \tag{24}$$

Taking expectation over the membrane fluctuations under the local analysis framework, where $\mathbf{y}_t$ is treated as locally fixed:

$$\mathbb{E}[\mathcal{L}(\hat{\mathbf{y}}_t)] = \mathcal{L}(\mathbf{y}_t^*) + \nabla\mathcal{L}(\mathbf{y}_t^*)^\top \mathbb{E}[\boldsymbol{\delta}\mathbf{y}_t] + \frac{1}{2} \mathbb{E}[\boldsymbol{\delta}\mathbf{y}_t^\top \nabla^2 \mathcal{L}(\mathbf{y}_t^*) \boldsymbol{\delta}\mathbf{y}_t] + O(\mathbb{E}[\|\boldsymbol{\delta}\mathbf{y}_t\|^3]) \tag{25}$$

Since $\mathbf{y}_t$ and $\bar{\mathbf{u}}_t$ are locally constant, and $\mathbb{E}[\varepsilon_t] = \mathbf{0}$ by definition of the mean-fluctuation decomposition:

$$\mathbb{E}[\boldsymbol{\delta}\mathbf{y}_t] = \mathbf{y}_t \odot g'(\bar{\mathbf{u}}_t) \odot \mathbb{E}[\varepsilon_t] + \frac{1}{2} \mathbf{y}_t \odot g''(\bar{\mathbf{u}}_t) \odot \mathbb{E}[\varepsilon_t^2] \tag{26}$$

$$= \frac{1}{2} \mathbf{y}_t \odot g''(\bar{\mathbf{u}}_t) \odot \mathbb{E}[\varepsilon_t^2] \tag{27}$$

For the second-order term, we assume independent fluctuations across dimensions with $\mathbb{E}[\varepsilon_{t,i}\varepsilon_{t,j}] = \delta_{ij}\sigma_i^2$, which is reasonable given the independent reset behavior in each dimension. The dominant contribution comes from the first-order term in $\boldsymbol{\delta y}_t$:

$$\mathbb{E}[\boldsymbol{\delta y}_t^\top \nabla^2 \mathcal{L}(\mathbf{y}_t^*)\boldsymbol{\delta y}_t] \approx \mathbb{E}[(\mathbf{y}_t \odot g'(\bar{\mathbf{u}}_t) \odot \boldsymbol{\varepsilon}_t)^\top \nabla^2 \mathcal{L}(\mathbf{y}_t^*)(\mathbf{y}_t \odot g'(\bar{\mathbf{u}}_t) \odot \boldsymbol{\varepsilon}_t)] \quad (28)$$

$$= \sum_i (y_{t,i}g'(\bar{u}_{t,i}))^2 [\nabla^2 \mathcal{L}(\mathbf{y}_t^*)]_{i,i}\sigma_i^2 \quad (29)$$

We define the regularization term to include both the bias correction and the quadratic regularization:

$$\mathcal{R}(\mathbf{y}_t, \bar{\mathbf{u}}_t) = \nabla \mathcal{L}(\mathbf{y}_t^*)^\top \left( \frac{1}{2}\mathbf{y}_t \odot g''(\bar{\mathbf{u}}_t) \odot (\sigma_1^2, \ldots, \sigma_d^2)^\top \right) + \frac{1}{2}\sum_i (y_{t,i}g'(\bar{u}_{t,i}))^2 [\nabla^2 \mathcal{L}(\mathbf{y}_t^*)]_{i,i}\sigma_i^2 \quad (30)$$

Therefore:

$$\mathbb{E}[\mathcal{L}(\hat{\mathbf{y}}_t)] = \mathcal{L}(\mathbf{y}_t \odot g(\bar{\mathbf{u}}_t)) + \mathcal{R}(\mathbf{y}_t, \bar{\mathbf{u}}_t) + \mathcal{O}(\|\boldsymbol{\varepsilon}_t\|^3) \quad (31)$$

## D.4 BOUNDEDNESS OF THE LIM LOSS

To establish the theoretical foundation for our regularization analysis, we first demonstrate the boundedness of $\mathbb{E}[\mathcal{L}(\hat{\mathbf{y}}_t)]$ by showing that both components of the gated output $\mathbf{y}_t \odot g(\mathbf{u}_t)$ remain bounded.

**Boundedness of $\mathbf{y}_t$ (Mamba output):** The HiPPO theory (Gu et al., 2020) ensures that Mamba's hidden state remains bounded through its initialization strategy. Specifically, HiPPO initializes the state matrix $\mathbf{A}$ with all negative real parts of eigenvalues, making the discretized system matrix $\overline{\mathbf{A}} = \exp(\Delta \mathbf{A})$ contractive with spectral radius less than 1. This contractivity property ensures that the recurrent dynamics (equation 3) cannot grow unboundedly. Even with Mamba's input-dependent discretization parameter $\Delta_t$, the constraint $\Delta_t > 0$ preserves the contractivity property since $\exp(\Delta_t \mathbf{A})$ maintains the same spectral properties as $\exp(\mathbf{A})$ when $\Delta_t > 0$. Therefore, for bounded inputs $\|\mathbf{x}_t\| \leq X_{\max}$, we have $\|\mathbf{h}_t\| \leq H$ for some constant $H$, which ensures $\|\mathbf{y}_t\| \leq M$ for some bound $M$.

**Boundedness of $g(\mathbf{u}_t)$:** The gating function is bounded by the design of our LIM mechanism. Since the membrane potential is bounded within $-M_u \leq \mathbf{u}_t \leq V_{th}$ as established in Section D.1, and the activation function $g(\cdot)$ (typically SiLU) is continuous on this compact interval, we have $|g(\mathbf{u}_t)| \leq G_{\max}$ for some constant $G_{\max}$.

**Final boundedness:** Since both components are bounded, the gated output satisfies $\|\mathbf{y}_t \odot g(\mathbf{u}_t)\| \leq M \cdot G_{\max}$. For a Lipschitz continuous loss function $\mathcal{L}$ with Lipschitz constant $L_{\text{lip}}$, we obtain:

$$\mathbb{E}[\mathcal{L}(\hat{\mathbf{y}}_t)] \leq \mathcal{L}(\mathbf{0}) + L_{\text{lip}} \cdot M \cdot G_{\max} \quad (32)$$

Therefore, the expected loss under our LIM mechanism is bounded, providing the theoretical foundation for our subsequent regularization analysis.

## D.5 BOUNDING THE REGULARIZATION TERM

We now establish the bound on the regularization term $\mathcal{R}(\mathbf{y}_t, \bar{\mathbf{u}}_t)$ as stated in Theorem 1.

From the analysis in the previous section, the dominant contribution to $\mathcal{R}(\mathbf{y}_t, \bar{\mathbf{u}}_t)$ comes from the quadratic term:

$$\mathcal{R}(\mathbf{y}_t, \bar{\mathbf{u}}_t) \approx \frac{1}{2}\sum_i (y_{t,i}g'(\bar{u}_{t,i}))^2 [\nabla^2 \mathcal{L}(\mathbf{y}_t^*)]_{i,i}\sigma_i^2 \quad (33)$$

Bounding the diagonal Hessian elements by the maximum eigenvalue $\lambda_{\max}$:

$$[\nabla^2 \mathcal{L}(\mathbf{y}_t^*)]_{i,i} \leq \lambda_{\max}(\nabla^2 \mathcal{L}(\mathbf{y}_t^*)) \quad (34)$$

Therefore:

$$\sum_i (y_{t,i}g'(\bar{u}_{t,i}))^2 [\nabla^2 \mathcal{L}(\mathbf{y}_t^*)]_{i,i}\sigma_i^2 \leq \lambda_{\max}(\nabla^2 \mathcal{L}(\mathbf{y}_t^*)) \sum_i (y_{t,i}g'(\bar{u}_{t,i}))^2 \sigma_i^2 \quad (35)$$

The term $\gamma$ in Theorem 1 captures the combined effect of model outputs, gate sensitivity, and fluctuation variance. We can bound:

$$\sum_i (y_{t,i} g'(\bar{u}_{t,i}))^2 \sigma_i^2 \leq \gamma \epsilon^2 \tag{36}$$

where $\gamma$ is a dimensionless constant and $\epsilon^2 = \mathbb{E}[\|\varepsilon_t\|^2] = \sum_i \sigma_i^2$.

Let $\lambda_{\max} = \max_t \lambda_{\max}(\nabla^2 \mathcal{L}(\mathbf{y}_t^*))$.

Therefore:

$$\mathcal{R}(\mathbf{y}_t, \bar{\mathbf{u}}_t) \leq \frac{1}{2}\lambda_{\max} \cdot \gamma \epsilon^2 = \frac{\gamma}{2} \cdot \lambda_{\max} \cdot \epsilon^2 \tag{37}$$

This establishes the bound stated in Theorem 1.

### D.6 Loss Landscape Visualization

To qualitatively verify our theorized regularization effect, Figure 7 presents a comparison of loss landscapes between Mamba fine-tuned with standard LoRA and our *Memba* approach. The landscapes are visualized for models fine-tuned on the Diabetic Retinopathy dataset from the VTAB-1k benchmark. In both plots, the red star indicates the original pre-trained model, while the cyan dot marks the converged solution after fine-tuning. The visualization reveals two key findings that align with our theoretical analysis: (1) *Memba* achieves a lower overall loss (minimum error 0.2459 vs. 0.2811), demonstrating its better optimization capability; and (2) *Memba* produces a smoother, more convex loss landscape with more gradual contour transitions. The smoother geometry suggests LIM helps avoid sharp minima, potentially contributing to its improved generalization performance and stability during fine-tuning.

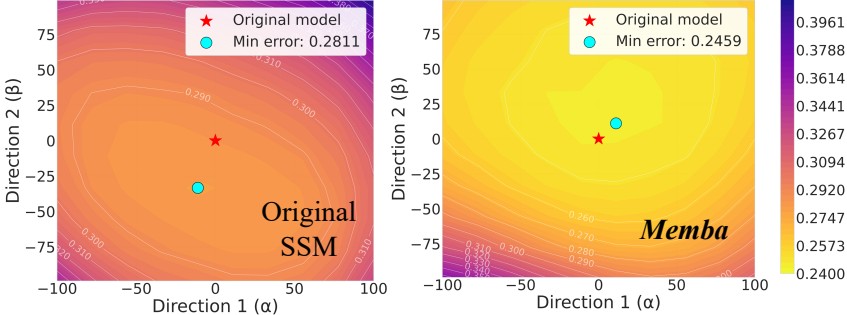

Figure 7: Loss landscapes of original SSM and *Memba*.

# E  HYPERPARAMETER DETAILS

## E.1  LANGUAGE TASK

For commonsense reasoning evaluation, we adopt the fine-tuning setup from (Hu et al., 2023; Yoshimura et al., 2024). We train on a combined commonsense reasoning dataset containing approximately 170K examples for 3 epochs with a batch size of 16. Our LoRA configuration applies rank-64 adapters to the projection components (`in_proj` and `out_proj`, denoted as $\mathbf{W}_{\text{in}}$ and $\mathbf{W}_{\text{out}}$), and rank-32 adapters to the gate components ($\mathbf{W}_{\text{in}}^{\text{gate}}$ and $\mathbf{W}_{\text{out}}^{\text{gate}}$). For the LIM neuron, we configure 4 chunks with a threshold of 1.0 and a leaky factor of 0.5. We employ model-specific learning rates: 1e-3 for *Memba*-130M, 5e-4 for *Memba*-370M and *Memba*-790M, and 1e-4 for *Memba*-1.4B. All models are trained using a linear learning rate scheduler with 100 warmup steps.

## E.2  VISION TASK

For vision evaluation, we use the VTAB-1K benchmark, which comprises 19 diverse datasets with 1,000 training examples each. Our training protocol consists of 100 epochs with a cosine learning rate scheduler, 10 epochs of warmup, a learning rate of 1e-3, weight decay of 1e-4, and batch size of 32. For parameter-efficient fine-tuning, we employ architecture-specific LoRA configurations. The Vim (Zhu et al., 2024) architecture uses ranks of 32 for $\mathbf{W}_{\text{in}}$, 96 for $\mathbf{W}_{\text{out}}$, and 16 for gate-related linear layers ($\mathbf{W}_{\text{in}}^{\text{gate}}$ and $\mathbf{W}_{\text{out}}^{\text{gate}}$). The VMamba (Liu et al., 2024c) architecture employs ranks of 64 for both $\mathbf{W}_{\text{in}}$ and $\mathbf{W}_{\text{out}}$, with rank 16 for gate components. For both architectures, we configure the LIM neuron with 4 temporal chunks, a threshold of 1.0, and a leaky factor of 0.5.

## E.3  S4 IMPLEMENTATION

We train the *Memba*-S4 architecture from scratch on the Long Range Arena (LRA) benchmarks (Tay et al., 2020), which include ListOps, Text, Retrieval, Image, and Pathfinder tasks, in Section G.4. These benchmarks are specifically designed to evaluate model performance on long-range sequence processing. Table 5 provides the experimental configuration for our S4 implementation. For the gate modification component, we employ a 32-rank for gate components ($\mathbf{W}_{\text{in}}^{\text{gate}}$ and $\mathbf{W}_{\text{out}}^{\text{gate}}$), with 4 chunks. Additionally, we set the LIM threshold to 1.0 and use a leaky factor of 0.5.

Table 5: Experimental settings on training S4 architecture. #layers: the number of layers. d_features: dimension of feature maps. d_state: dimension of state. lr: learning rate. we adopt the S5 (Smith et al., 2022) training setup for other hyperparameters.

|            | ListOps | Text  | Retrieval | Image | Pathfinder |
|------------|---------|-------|-----------|-------|------------|
| #layers    | 8       | 8     | 6         | 6     | 6          |
| d_feature  | 128     | 128   | 128       | 512   | 256        |
| d_state    | 64      | 4     | 4         | 64    | 64         |
| batch size | 50      | 16    | 32        | 50    | 64         |
| lr         | 0.003   | 0.003 | 0.002     | 0.005 | 0.005      |
| epoch      | 40      | 50    | 30        | 250   | 200        |

# F  INDIVIDUAL TASK ACCURACY FOR VTAB-1K BENCHMARK

We provide the details of individual task performance on VTAB-1K benchmark as shown in Table 6. The Table 3 shows the averaged values of each category (Natural, Specialzed, and Structured).

Table 6: Per-task performances of VTAB-1k benchmark in Table 3. We separate three tables according to baseline architecture: ViT-S, Vim-S, and Vanilla-VMamba-S. *Avg.* and *Overall Avg.* represent average accuracy of each category and the average accuracy of overall tasks. **Bold** and underlined values represent the best and second-best performance respectively.

| Method | Natural | | | | | | | | Specialized | | | | | Structured | | | | | | | | | Overall Avg. |
|---|---|---|---|---|---|---|---|---|---|---|---|---|---|---|---|---|---|---|---|---|---|---|---|
| | CIFAR-100 | Caltech101 | DTD | Flowers102 | Pets | SVHN | Sun397 | Avg. | Camelyon | EuroSAT | Resisc45 | Retinopathy | Avg. | Clev-Count | Clev-Dist | DMLab | KITTI-Dist | dSpr-Loc | dSpr-Ori | sNORB-Azim | sNORB-Elev | Avg. | |
| Scracth | 4.9 | 11.1 | 9.7 | 24.4 | 3.4 | 19.4 | 1.6 | 10.7 | 65.0 | 57.3 | 28.6 | 73.6 | 56.1 | 20.9 | 48.1 | 26.2 | 45.0 | 7.3 | 27.1 | 6.2 | 17.8 | 24.8 | 26.2 |
| Full | 30.4 | 69.3 | 37.6 | 65.8 | 50.2 | 87.8 | 21.5 | 51.8 | 76.2 | 85.2 | 66.8 | 63.0 | 72.8 | 33.5 | 56.8 | 40.3 | 66.8 | 73.5 | 40.4 | 28.0 | 22.8 | 45.3 | 53.5 |
| LoRA | 55.3 | 87.6 | 67.3 | 90.2 | 85.1 | | 39.6 | 73.6 | 81.7 | 94.1 | 80.5 | 72.7 | 82.2 | 77.6 | 59.1 | 47.0 | 81.6 | 81.5 | 49.6 | 29.4 | 35.1 | 57.6 | 68.7 |
| Adaptformer | 56.0 | 88.0 | 65.6 | 90.5 | 90.5 | 85.3 | 39.5 | 73.6 | 84.0 | 93.8 | 82.2 | 72.7 | 83.2 | 77.5 | 59.0 | 46.7 | 79.3 | 83.7 | 50.4 | 30.5 | 35.4 | 57.8 | 69.0 |
| Adapter+ | 56.6 | 89.2 | 66.8 | 91.1 | 90.2 | 88.2 | 40.7 | 74.7 | 84.8 | 94.2 | 82.9 | 72.4 | 83.6 | 76.8 | 59.7 | 48.4 | 80.5 | 87.8 | 51.9 | 32.4 | 33.1 | 58.8 | 69.9 |

(a) ViT-S architecture pre-trained with ImageNet-1K.

| Method | Natural | | | | | | | | Specialized | | | | | Structured | | | | | | | | | Overall Mean |
|---|---|---|---|---|---|---|---|---|---|---|---|---|---|---|---|---|---|---|---|---|---|---|---|
| | CIFAR-100 | Caltech101 | DTD | Flowers102 | Pets | SVHN | Sun397 | Mean | Camelyon | EuroSAT | Resisc45 | Retinopathy | Mean | Clev-Count | Clev-Dist | DMLab | KITTI-Dist | dSpr-Loc | dSpr-Ori | sNORB-Azim | sNORB-Elev | Mean | |
| Scratch | 5.6 | 11.9 | 5.9 | 12.1 | 5.0 | 16.3 | 1.6 | 8.3 | 61.6 | 62.3 | 13.8 | 61.6 | 49.8 | 28.9 | 53.2 | 22.5 | 40.9 | 38.6 | 11.8 | 11.3 | 18.3 | 28.2 | 25.4 |
| Full | 51.6 | 83.0 | 61.5 | 71.1 | 45.0 | 88.4 | 14.9 | 59.4 | 66.5 | 88.1 | 63.3 | 57.1 | 68.7 | 48.8 | 60.4 | 38.8 | 55.0 | 6.3 | 11.8 | 30.0 | 24.1 | 34.4 | 50.8 |
| LoRA(embed) | 51.7 | 84.9 | 59.2 | 75.5 | 88.0 | 53.4 | 39.8 | 64.7 | 77.5 | 91.2 | 71.5 | 70.0 | 77.5 | 42.9 | 52.0 | 54.9 | 73.0 | 54.9 | 47.0 | 19.7 | 26.2 | 43.8 | 58.6 |
| LoRA(x_proj) | 59.2 | 88.0 | 67.5 | 88.9 | 90.4 | 83.9 | 43.0 | 74.4 | 82.7 | 93.9 | 80.3 | 70.7 | 81.9 | 72.5 | 60.7 | 43.6 | 80.6 | 75.0 | 45.8 | 27.8 | 33.1 | 54.9 | 67.8 |
| LoRA(dt_proj) | 61.6 | 90.1 | 66.7 | 89.0 | 90.7 | 86.1 | 43.4 | 75.4 | 83.2 | 94.9 | 81.2 | 72.9 | 83.1 | 76.9 | 59.8 | 47.0 | 79.8 | 75.6 | 50.7 | 30.3 | 37.0 | 57.1 | 69.3 |
| LoRA(out_proj) | 64.2 | 89.8 | 68.6 | 91.4 | 91.2 | 86.0 | 43.8 | 76.4 | 84.9 | 94.9 | 83.4 | 72.7 | 84.0 | 84.5 | 62.7 | 49.5 | 81.3 | 77.1 | 52.0 | 32.8 | 40.9 | 60.1 | 71.1 |
| LoRA(in_proj) | 63.4 | 90.3 | 68.2 | 91.0 | 91.0 | 88.6 | 43.5 | 76.6 | 84.7 | 95.0 | 83.8 | 72.9 | 84.1 | 84.0 | 61.3 | 48.6 | 80.0 | 83.5 | 52.6 | 31.8 | 39.5 | 60.2 | 71.3 |
| LoRA$_p$(Z) | 61.7 | 90.6 | 68.1 | 90.3 | 90.7 | 88.4 | 43.2 | 76.2 | 85.6 | 95.2 | 83.5 | 72.8 | 84.3 | 82.4 | 60.6 | 48.2 | 81.9 | 81.6 | 52.4 | 31.4 | 39.5 | 59.7 | 70.9 |
| LoRA$_p$(X) | 63.3 | 89.3 | 69.4 | 90.6 | 91.5 | 88.9 | 43.6 | 76.6 | 84.5 | 95.3 | 83.4 | 72.4 | 83.9 | 84.9 | 62.9 | 48.6 | 81.4 | 82.8 | 52.7 | 33.1 | 40.4 | 60.8 | 71.5 |
| Hybrid | 63.7 | 90.2 | 69.4 | 90.9 | 90.1 | 90.1 | 43.9 | 77.0 | 86.3 | 95.2 | 83.9 | 72.2 | 84.4 | 84.6 | 61.7 | 50.0 | 81.0 | 86.4 | 54.0 | 34.6 | 40.2 | 61.6 | 72.1 |
| *Memba*(in_proj) | 63.4 | 91.0 | 69.2 | 90.6 | 91.4 | 89.2 | 43.6 | 76.9 | 86.2 | 95.5 | 83.6 | 75.1 | 85.1 | 83.6 | 62.3 | 49.5 | 82.3 | 83.8 | 52.7 | 32.8 | 38.8 | 60.7 | 71.8 |
| *Memba*(out_proj) | 63.1 | 91.4 | 68.8 | 91.4 | 91.5 | 88.4 | 43.9 | 76.9 | 86.6 | 95.3 | 84.1 | 75.3 | 85.3 | 84.6 | 63.2 | 50.3 | 82.7 | 82.8 | 52.4 | 33.4 | 40.1 | 61.2 | 72.1 |
| *Memba*(in+out_proj) | 63.2 | 91.1 | 68.6 | 91.3 | 91.4 | 89.9 | 44.0 | 77.1 | 87.2 | 95.4 | 84.5 | 75.5 | 85.7 | 84.8 | 62.4 | 50.4 | 83.3 | 85.1 | 52.5 | 34.7 | 40.3 | 61.7 | 72.4 |

(b) Vim-S architecture pre-trained with ImageNet-1K.

| Method | Natural | | | | | | | | Specialized | | | | | Structured | | | | | | | | | Overall Mean |
|---|---|---|---|---|---|---|---|---|---|---|---|---|---|---|---|---|---|---|---|---|---|---|---|
| | CIFAR-100 | Caltech101 | DTD | Flowers102 | Pets | SVHN | Sun397 | Mean | Camelyon | EuroSAT | Resisc45 | Retinopathy | Mean | Clev-Count | Clev-Dist | DMLab | KITTI-Dist | dSpr-Loc | dSpr-Ori | sNORB-Azim | sNORB-Elev | Mean | |
| LoRA(in_proj) | 62.3 | 89.7 | 70.4 | 92.3 | 92.7 | 92.5 | 44.5 | 77.8 | 86.7 | 95.6 | 85.3 | 76.7 | 86.0 | 85.3 | 64.1 | 54.1 | 85.5 | 86.5 | 54.5 | 35.1 | 42.4 | 63.4 | 73.5 |
| LoRA(out_proj) | 62.3 | 89.8 | 70.2 | 92.0 | 92.5 | 91.7 | 43.7 | 77.4 | 87.2 | 95.2 | 85.5 | 76.3 | 86.1 | 90.1 | 62.5 | 54.7 | 83.7 | 88.3 | 56.0 | 37.4 | 41.9 | 64.3 | 73.7 |
| LoRA(in+out_proj) | 61.0 | 89.7 | 69.1 | 93.0 | 92.8 | 91.8 | 43.8 | 77.3 | 86.8 | 94.9 | 85.0 | 76.6 | 85.8 | 83.7 | 62.5 | 54.7 | 84.4 | 90.8 | 55.0 | 35.9 | 39.5 | 63.3 | 73.2 |
| *Memba*(in_proj) | 62.3 | 89.9 | 70.4 | 92.7 | 93.0 | 91.5 | 43.9 | 77.7 | 87.0 | 95.3 | 85.4 | 76.5 | 86.1 | 88.1 | 64.0 | 53.6 | 84.1 | 89.3 | 55.3 | 34.1 | 42.8 | 63.9 | 73.7 |
| *Memba*(out_proj) | 62.6 | 90.1 | 71.2 | 92.4 | 92.5 | 91.8 | 43.8 | 77.8 | 87.4 | 95.1 | 85.1 | 76.2 | 86.0 | 90.6 | 63.8 | 55.4 | 83.8 | 91.5 | 56.2 | 37.2 | 40.5 | 64.9 | 74.1 |
| *Memba*(in+out_proj) | 61.1 | 87.8 | 68.4 | 93.2 | 92.2 | 92.8 | 44.5 | 77.1 | 87.9 | 95.0 | 85.1 | 76.7 | 86.1 | 88.1 | 63.9 | 54.6 | 83.7 | 90.4 | 54.1 | 35.9 | 40.8 | 63.9 | 73.5 |

(c) Vanilla-VMamba-S architecture pre-trained with ImageNet-1K.

# G  ADDITIONAL ABLATION STUDY

In this section, we conduct comprehensive ablation studies to provide deeper insights into *Memba*'s design choices and performance characteristics. We systematically examine five key aspects: (1) the impact of chunk count on LIM neuron performance, (2) accuracy comparisons under iso-parameter conditions, (3) parameter allocation within the LIM neuron, (4) performance comparison with traditional recurrent gating mechanisms, and (5) computational overhead analysis through inference time measurements.

## G.1  NUMBER OF CHUNKS

The number of chunks in the LIM neuron determines the computational iteration of membrane dynamics. Table 7 presents accuracy comparisons across different numbers of chunks for both language and vision tasks. For language tasks, we employ the *Memba*-130M architecture on commonsense reasoning benchmarks. For vision tasks, we utilize the Vim-S architecture evaluated on Camelyon, EuroSAT, Resisc45, and Retinopathy datasets from the VTAB-1K benchmark. Across both task domains, LIM with 4 chunks achieves optimal performance while maintaining reasonable

inference time. The reported inference time corresponds to LIM neuron processing only. Notably, inference time does not scale proportionally with the number of chunks, as increasing the chunk count results in smaller sequence lengths per chunk.

Table 7: Accuracy comparison across different numbers of chunks. #chunk represents the number of chunks in the LIM neuron. For language and vision tasks, we use ***Memba***-130M and Vim-S architectures, respectively. Inference times are measured during vision task evaluation.

| #chunk | Language (%) | Vision (%) | Inference time (ms) |
|--------|--------------|------------|---------------------|
| 2 | 43.9 | 85.60 | 1.5 |
| 4 | 44.3 | 85.66 | 1.7 |
| 6 | 44.1 | 85.43 | 1.8 |
| 8 | 43.7 | 85.36 | 2.0 |

## G.2 ISO-PARAMETER ACCURACY COMPARISON

To verify the effectiveness of ***Memba*** under fair comparison conditions, we evaluate accuracy with the same number of learnable parameters as previous works, particularly MambaPeft (Yoshimura et al., 2024), as shown in Table 8. To match the parameter count, we apply LoRA only to the output projection ($W_{out}$) and reduce the rank of gate components ($W_{in}^{gate}$ and $W_{out}^{gate}$) to 16. This configuration matches the parameter count of MambaPeft's "LoRA (in_proj)" setting, and our ***Memba*** consistently achieves better performance across all architecture sizes.

Table 8: Performance comparison between ***Memba*** and MambaPeft (Yoshimura et al., 2024) on commonsense reasoning datasets. Reported values are accuracy percentages (%). **Bold** and underlined entries indicate the best and second-best performance, respectively. HS and WG refer to the HellaSwag and WinoGrande datasets.

| Model | Method | #Params(%) | BoolQ | PIQA | SIQA | HS | WG | ARC-e | ARC-c | OBQA | Avg. |
|-------|--------|-----------|-------|------|------|-----|-----|-------|-------|------|------|
| | Full | 100 | 56.1 | 65.3 | 38.7 | 35.3 | 52.0 | 46.4 | 25.7 | 32.8 | 43.8 |
| Mamba 130M | Additional-scan | 0.51 | 57.8 | 64.1 | 37.5 | 34.5 | 53.0 | 41.3 | 23.5 | 30.0 | 42.7 |
| | Affix-tuning | 64.64 | 59.7 | 64.3 | 38.2 | **35.2** | 51.9 | 42.9 | 24.0 | 29.0 | 43.2 |
| | LoRA (in_proj) | 2.23 | 53.5 | 62.9 | 38.2 | 33.8 | **53.1** | 46.4 | 23.7 | **30.8** | 42.8 |
| | LoRA$_p$ (X) | 2.67 | **61.7** | 64.0 | **39.5** | 34.3 | 52.2 | 43.5 | **25.3** | 29.4 | **43.7** |
| | ***Memba*** | 2.23 | 56.3 | **64.5** | 39.1 | 34.4 | 51.6 | **48.0** | 24.0 | 29.4 | 43.4 |
| | Full | 100 | 58.1 | 69.9 | 41.9 | 45.7 | 53.8 | 52.7 | 29.7 | 33.4 | 48.2 |
| Mamba 370M | Additional-scan | 0.47 | **61.9** | **69.3** | 41.2 | 45.3 | 54.9 | 49.5 | **28.4** | 31.4 | 47.7 |
| | Affix-tuning | 68.88 | 61.2 | 68.4 | 39.6 | **46.2** | 55.4 | 48.2 | 28.2 | 30.6 | 47.2 |
| | LoRA (in_proj) | 2.07 | 55.4 | 68.6 | 41.0 | 44.7 | 54.1 | 52.4 | 28.3 | **33.4** | 47.2 |
| | LoRA$_p$ (X) | 2.67 | 60.8 | 68.8 | **42.1** | 44.7 | **56.2** | 50.4 | 27.4 | 32.2 | 47.8 |
| | ***Memba*** | 2.07 | 56.2 | 69.2 | **42.1** | 44.8 | 55.7 | 56.0 | 28.3 | **33.4** | 48.2 |
| | Full | 100 | 62.0 | 72.1 | 44.8 | 54.0 | 55.9 | 57.7 | 31.2 | 35.2 | 51.6 |
| Mamba 790M | Additional-scan | 0.33 | **63.0** | 71.9 | 41.9 | 54.2 | 57.1 | 54.9 | 30.0 | 32.6 | 50.7 |
| | Affix-tuning | 69.99 | 61.0 | **72.5** | 41.0 | **54.9** | 55.6 | 54.6 | 29.6 | 33.8 | 50.4 |
| | LoRA (in_proj) | 1.47 | 61.7 | 71.9 | 44.0 | 50.8 | 56.7 | 56.3 | 30.5 | 33.8 | 50.7 |
| | LoRA$_p$ (X) | 1.75 | 59.9 | 72.2 | **44.2** | 52.8 | **58.0** | 53.7 | 30.8 | **34.8** | 50.8 |
| | ***Memba*** | 1.47 | 58.8 | 72.2 | 44.1 | 54.7 | **58.3** | 61.5 | 31.1 | 34.0 | **51.8** |
| | Full | 100 | 61.4 | 73.3 | 43.9 | 56.9 | 59.0 | 59.7 | 34.0 | 35.4 | 53.0 |
| Mamba 1.4B | Additional-scan | 0.26 | 63.0 | 73.5 | 42.8 | 57.5 | 60.5 | 60.9 | 32.4 | 37.4 | 53.5 |
| | LoRA (in_proj) | 1.13 | 62.6 | 73.6 | **43.7** | 55.6 | 59.7 | 58.3 | 31.7 | 35.6 | 52.6 |
| | LoRA$_p$ (X) | 1.36 | 63.1 | 73.5 | 42.7 | 57.7 | **61.6** | 60.4 | **32.9** | 37.4 | 53.7 |
| | ***Memba*** | 1.13 | **64.1** | **73.9** | 43.4 | **58.3** | 60.8 | **65.1** | 32.3 | **37.6** | **54.5** |

## G.3 PARAMETER ALLOCATION ANALYSIS

To provide deeper insights into ***Memba***'s parameter efficiency, Table 9 presents a detailed breakdown of parameter allocation across different model sizes. Unlike traditional PEFT methods that distribute parameters across various projection layers, ***Memba*** strategically allocates all trainable parameters to the gate projections ($W_{in}^{gate}$ and $W_{out}^{gate}$) within the LIM neuron. For instance, in the Mamba-790M configuration, ***Memba*** allocates 4.72M parameters to input gate projections and 7.08M parameters

to output gate projections, totaling 11.80M parameters, identical to the baseline LoRA (in_proj) method. This targeted allocation enables *Memba* to enhance temporal processing capabilities while maintaining parameter efficiency. Notably, *Memba* consistently outperforms baseline methods across all model sizes: achieving 1.1% improvement over LoRA_p(X) for Mamba-790M (51.8% vs 50.8%) and 0.8% improvement for Mamba-1.4B (54.5% vs 53.7%), demonstrating that strategic parameter placement in the gating mechanism is more effective than conventional projection-based adaptations.

Table 9: Parameter allocation and performance comparison for *Memba*. The "#Params" column represents the total number of trainable parameters, "$\mathbf{W}_{in}^{gate} + \mathbf{W}_{out}^{gate}$" shows the parameters dedicated to the gate modules in the LIM neuron, and "Others" represents all remaining parameters excluding the gate modules.

| Model | Method | # Params | $\mathbf{W}_{in}^{gate} + \mathbf{W}_{out}^{gate}$ | Others | Avg. Acc. (%) |
|-------|--------|----------|-----------|--------|---------------|
| Mamba-130M | LoRA (in_proj) | 2.95M | - | - | 42.8 |
| | LoRA_p(X) | 3.54M | - | - | 43.7 |
| | *Memba* | **2.95M** | **1.18M** | **1.77M** | **43.4** |
| Mamba-370M | LoRA (in_proj) | 7.87M | - | - | 47.2 |
| | LoRA_p(X) | 9.44M | - | - | 47.8 |
| | *Memba* | **7.87M** | **3.15M** | **4.72M** | **48.2** |
| Mamba-790M | LoRA (in_proj) | 11.80M | - | - | 50.7 |
| | LoRA_p(X) | 14.16M | - | - | 50.8 |
| | *Memba* | **11.80M** | **4.72M** | **7.08M** | **51.8** |
| Mamba-1.4B | LoRA (in_proj) | 15.73M | - | - | 52.6 |
| | LoRA_p(X) | 18.87M | - | - | 53.7 |
| | *Memba* | **15.73M** | **6.29M** | **9.44M** | **54.5** |

## G.4 COMPARISON WITH TRADITIONAL RECURRENT GATING MECHANISMS

Our LIM neuron is designed to enhance Mamba's temporal adaptation capabilities through recurrently evolving membrane potentials. To evaluate this approach against established alternatives, we compare LIM with traditional recurrent architectures by replacing LIM with LSTM (Hochreiter & Schmidhuber, 1997) and GRU (Chung et al., 2014) in the gate path.

Table 10 presents a comprehensive comparison across two distinct scenarios: vision adaptation tasks using the Vim-S architecture on VTAB-1k, and sequence modeling tasks on the Long Range Arena (LRA) benchmark (Tay et al., 2020) using S4 (Gu et al., 2021a) trained from scratch. Our LIM neuron consistently outperforms both LSTM and GRU across domains while offering two significant advantages: (1) zero learnable parameters and (2) lower inference latency. The superior performance of LIM stems from its temporal processing driven by membrane dynamics, whereas LSTM and GRU rely on parameter-heavy gating units that increase computational cost with trivial improvement to temporal modeling capability.

Table 10: Comparison of LIM with other recurrent units in terms of performance, parameter count, and latency. We report average accuracy (%) on two benchmarks: VTAB (using the Vim-S architecture) and Long Range Arena (using the S4 architecture). # Params denotes the number of learnable parameters per unit given a hidden dimension $H$, excluding the input and output projection layers ($\mathbf{W}_{in}^{gate}$ and $\mathbf{W}_{out}^{gate}$) which are applied consistently across all methods. Latency refers to inference time per batch measured on the S4 architecture using an A100 GPU.

| Gate | # Params | Vim-S | S4 | | | | | | Latency (s) |
|------|----------|-------|--------|------|-----------|-------|------------|------|-------------|
| | | VTAB | ListOps | Text | Retrieval | Image | Pathfinder | Avg. | |
| LSTM | $8H^2 + 4H$ | 71.59 | **62.10** | 88.00 | 91.37 | 89.60 | 96.75 | 85.57 | 0.189 |
| GRU | $6H^2 + 3H$ | 71.68 | 61.80 | 87.97 | **91.70** | 89.02 | 96.81 | 85.46 | 0.188 |
| LIM (ours) | **0** | **72.33** | 62.05 | **89.60** | 91.52 | **89.88** | **97.12** | **86.04** | **0.150** |

## G.5 INFERENCE TIME ANALYSIS

The proposed *Memba* incorporates the LIM neuron, which introduces inevitable recurrent computation for membrane accumulation and reset mechanisms. To quantify this computational overhead

compared to previous PEFT methods on Mamba, we provide an inference time comparison based on Table 4(a). Table 11 demonstrates how the LIM module affects both accuracy and inference time. Since membrane transfer has a negligible impact on inference time, we exclude its influence from our analysis. We conduct experiments on language and vision tasks using the ***Memba***-130M and Vim-S architecture, with inference times measured on single-batch evaluation.

***Memba***, which integrates LIM, LoRA, and membrane transfer, achieves higher performance on both language and vision tasks. However, this comes with an 8.8% inference time overhead compared to Mamba with LoRA. This overhead stems from the iterative operations required for membrane accumulation in the LIM neuron. Nevertheless, the 8.8% increase represents a modest computational cost, and our chunked sequence processing approach in the LIM neuron effectively balances performance gains with inference efficiency.

Table 11: Inference time analysis with different ***Memba*** components. The table shows the impact of each component on accuracy and computational overhead. For language and vision tasks, we use ***Memba***-130M and Vim-S architectures, respectively. Inference times are measured during vision task evaluation on a single batch.

| ① LIM | ② LoRA | ③ Membrane Transfer | Language (%) | Vision (%) | Inference Time (s) |
|---|---|---|---|---|---|
| ✗ | ✓ | ✗ | 43.8 | 85.3 | 0.517 |
| ✓ | ✓ | ✓ | 44.3 | 85.7 | 0.563 |

## H   COMPREHENSIVE MEMORY AND LATENCY ANALYSIS

To provide a complete understanding of the computational requirements of ***Memba***, we present comprehensive GPU memory and inference latency comparisons across baseline methods, model sizes, and task domains. For language tasks, we measure GPU peak memory during fine-tuning and per-sample inference latency on the ARC-Easy benchmark, shown in Table 12.

Table 12: GPU memory and inference latency comparison on language tasks (ARC-Easy benchmark). Latency represents per-sample inference time.

| Model | Method | Learnable Param (%) | GPU Peak Memory (GB) | Latency (s) |
|---|---|---|---|---|
| Mamba-130M | SLL LoRA | 1.45 | 2.62 | 0.015 |
| | Additional-scan | 0.51 | 2.17 | 0.014 |
| | Affix-tuning | 0.17 | 2.49 | 0.013 |
| | LoRA (in+out) | 3.53 | 2.51 | 0.014 |
| | ***Memba*** | 3.95 | 2.82 | 0.020 |
| Mamba-370M | SLL LoRA | 2.30 | 5.00 | 0.027 |
| | Additional-scan | 0.47 | 4.15 | 0.023 |
| | Affix-tuning | 0.16 | 4.50 | 0.021 |
| | LoRA (in+out) | 2.28 | 5.00 | 0.024 |
| | ***Memba*** | 3.67 | 5.80 | 0.044 |
| Mamba-790M | SLL LoRA | 3.10 | 8.20 | 0.028 |
| | Additional-scan | 0.33 | 6.56 | 0.023 |
| | Affix-tuning | 0.11 | 6.97 | 0.021 |
| | LoRA (in+out) | 2.32 | 8.03 | 0.024 |
| | ***Memba*** | 2.61 | 9.21 | 0.045 |
| Mamba-1.4B | SLL LoRA | 4.64 | 12.49 | 0.028 |
| | Additional-scan | 0.26 | 9.65 | 0.024 |
| | Affix-tuning | 0.09 | 9.71 | 0.021 |
| | LoRA (in+out) | 1.80 | 11.90 | 0.025 |
| | ***Memba*** | 2.02 | 13.60 | 0.045 |

***Memba*** incurs approximately 12-14% additional GPU memory overhead compared to LoRA (in+out), which is required for storing membrane potentials across layers to enable cross-layer membrane transfer. We observe that inference latency appears to be largely independent of hidden dimension size. Mamba-370M, 790M, and 1.4B exhibit similar latency despite having different hidden dimensions, as they share the same number of layers. In contrast, Mamba-130M shows noticeably lower latency due to its shallower architecture with fewer layers. Regarding the latency overhead introduced by

recurrent operations in the LIM neuron, as discussed in the Limitation section (Section 6), this latency can be substantially reduced through CUDA kernel optimization.

Table 13: GPU memory and inference latency comparison on vision tasks (Caltech101 dataset in VTAB-1k benchmark).

| Model | Method | Learnable Param (%) | GPU Peak Memory (GB) | Latency (s) |
|-------|--------|---------------------|----------------------|-------------|
| Vim-S | LoRA (in+out) | 13.98 | 7.68 | 0.517 |
|       | *Memba* | 15.60 | 9.85 | 0.563 |

For vision tasks in Table 13, we observe similar patterns on the VTAB-1k benchmark using the Vim-S architecture. The memory overhead is approximately doubled compared to language tasks because Vim-S uses bidirectional SSM with two gate paths, requiring storage of membrane potentials for two LIM neurons. The inference latency increase remains modest. These resource requirements represent a reasonable trade-off given the substantial performance improvements demonstrated in Tables 2 and 3.

## I  VISUALIZATION OF TEMPORAL PROCESSING

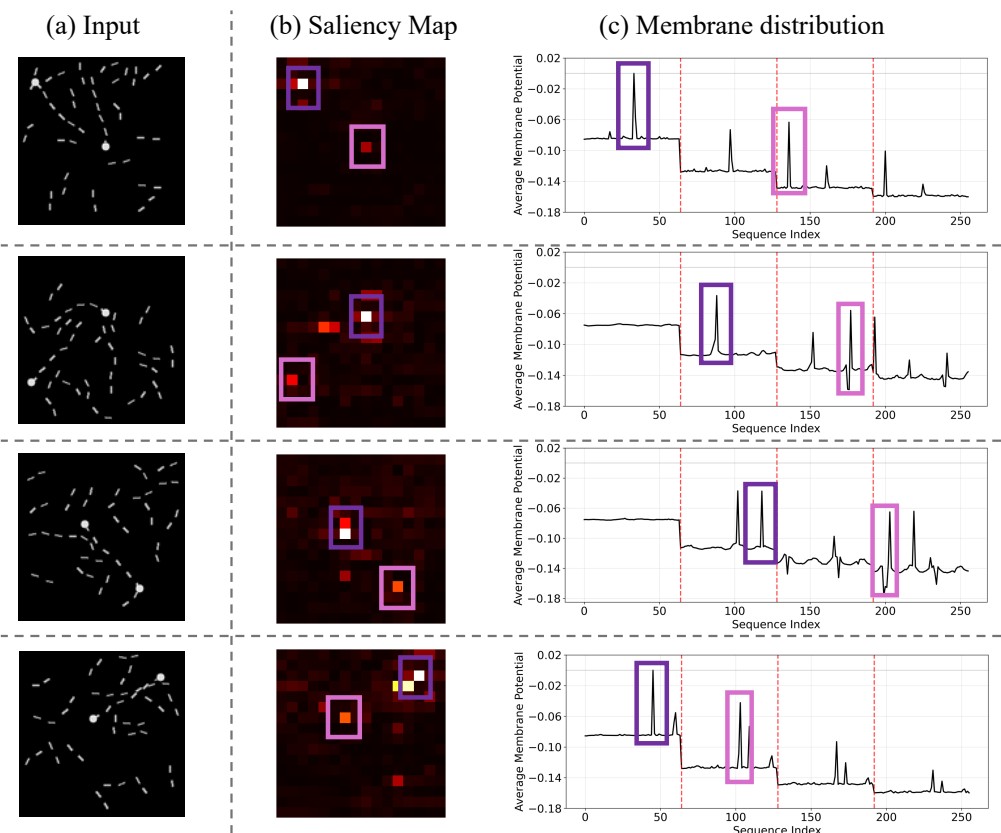

Figure 8: Membrane-driven temporal processing in *Memba*. Given input image (a), we present saliency maps (b) and associated membrane distributions (c) throughout the sequence processing.

To understand *Memba*'s temporal processing, we analyze membrane dynamics on the Pathfinder task. Figure 8 shows the relationship between visual attention and membrane activity during sequence processing. Task-critical path segments, highlighted in purple and pink, trigger pronounced membrane responses, demonstrating how the LIM neuron selectively amplifies relevant visual elements. Additionally, the membrane baseline declines across temporal chunks, indicating adaptive forgetting as new information accumulates.

