# OpenReview forum: "Memba: Membrane-driven Parameter-Efficient Fine-Tuning for Mamba"
_ICLR.cc/2026/Conference — ICLR 2026 Poster_

### Official Review · Reviewer_VB1b · 2025-10-27

**Soundness:** 3
**Presentation:** 3
**Contribution:** 2
**Rating:** 4
**Confidence:** 4

**Summary:**

This paper introduces Memba, which combines LIM with LoRA and cross-layer membrane transfer, aiming to improve parameter-efficient adaptation. The proposed Memba demonstrates superior performance over existing PEFT methods on language (commonsense reasoning) and vision (VTAB-1k) tasks, emphasizing parameter efficiency and temporal modeling capabilities.

**Strengths:**

The membrane-driven PEFT method is conceptually novel, as it creatively integrates LIF modules into SSM fine-tuning, addressing a gap in temporal adaptation for Mamba models.

The chunked processing and cross-layer transfer efficiently handle long sequences without adding learnable parameters.

**Weaknesses:**

This work is actually a generic PEFT method, not exclusive to Mamba, as it involves adding LIM to LoRA's hidden layers. Therefore, I would like to see experimental results on Transformers.

I find the absence of comparisons with advanced PEFT methods for Transformers (e.g., recent adapters or prompt-based techniques) or other SSM variants (e.g., Mamba-Adapter) reduces the claim of general superiority.

While Memba improves performance, the 8.8% inference overhead (Table 11) is non-trivial for real-time applications, and the paper underdiscusses trade-offs, such as energy efficiency or scalability on edge devices. Notably, spike networks (SNNs) are primarily beneficial for their low energy consumption, but the motivation for using LIM in this work does not seem to be energy efficiency. Therefore, I am puzzled as to why the authors chose to use LIM instead of other RNN-style modules, which would seem to be a more straightforward choice.

The LIM neuron is not entirely similar to the LIF used in SNNs, as it discards binary outputs. In this regard, this module actually shows similarities to traditional RNN gates, but the ablation (Table 10) only briefly compares them. A more rigorous analysis of novelty versus established gating mechanisms is needed.



The vision tasks using VTAB-1k fail to demonstrate the performance superiority, as previous work has shown that different PEFT methods perform comparably on this dataset [1]. Therefore, semantic segmentation and object detection tasks are imperative.

GPU memory and latency comparisons among different methods are missing.

Refs:
[1] Lessons and Insights from a Unifying Study of Parameter-Efficient Fine-Tuning (PEFT) in Visual Recognition. CVPR 2025.

**Questions:**

Please see weaknesses.

---

> ### Author Response · Authors · 2025-11-21
>
> We greatly appreciate your efforts in reviewing our work and providing valuable feedback, which encourages us to improve our work.
>
> > **Weakness #1**
>
> We deeply appreciate your concern about our motivation and respectfully argue that Memba is not a generic PEFT method. It is explicitly designed to solve Mamba-specific challenges. As stated in our introduction (Lines 28-40), our work is motivated by the failure of Transformer-based PEFT methods when applied to Mamba. Mamba's architecture differs fundamentally from Transformers in two critical ways: (1) it employs a simplified gating mechanism lacking the sophisticated gating structures like LSTM and GRU, and (2) its core SSM components are sensitive to parameter modifications, with recent studies [2, 3] demonstrating that directly fine-tuning these components degrades performance.
>
> Memba is designed specifically to address these Mamba-specific problems. We freeze the sensitive SSM core and introduce a parameter-free temporal adaptation module (LIM) exclusively into the gating path. Furthermore, the temporal properties of our LIM neuron are inherently well-aligned with SSM's sequential state evolution, enabling it to incorporate temporal adaptation during fine-tuning without disrupting the balanced dynamics of pre-trained SSMs.
>
> Applying LIM to a Transformer would be fundamentally different because: (1) Transformers use attention mechanisms rather than recurrent state evolution, (2) Transformer gates (in feedforward layers) don't have the same temporal dependencies that our membrane dynamics exploit. We will sharpen this Mamba-specific motivation in the revised introduction to make our design rationale clearer.
>
> > **Weakness #2**
>
> Thank you for raising this concern and pointing us to relevant concurrent work, particularly Mamba-Adapter [4]. We first want to clarify that our main comparison baselines are Mamba-based architectures, as our method is specifically designed for SSMs. We compare against the most comprehensive prior work on Mamba PEFT: MambaPEFT [2], SLL LoRA [3], and various LoRA configurations.
>
> Regarding Mamba-Adapter [4], we would like to clarify why their results are not directly comparable to ours, despite some dataset overlap. While they evaluate on datasets like CIFAR-100 and SVHN, the experimental settings are fundamentally different. Our vision results in Table 3 are on the standardized VTAB-1k benchmark, a low-data regime where each task is trained on 1,000 samples. However, the results in Mamba-Adaptor (Table 4 in Mamba-Adaptor paper) are on the full dataset, such as CIFAR-100 with 50,000 training samples and Food101 with 75,000+ samples. The much higher absolute performance numbers in Mamba-Adapter are expected results of using 50$\times$ more training data. The VTAB-1k benchmark is specifically designed to evaluate parameter-efficient fine-tuning in low-data scenarios, which is the primary use case for PEFT methods. Furthermore, the code for Mamba-Adapter is not publicly released, preventing us from conducting experiments under their settings for fair comparison.
>
> > **Weakness 3 & 4**
>
> We appreciate your concerns about the 8.8% inference overhead and the motivation for using LIM. The LIM neuron is inspired by the Leaky Integrate-and-Fire (LIF) neuron, but we have modified the reset function to produce floating-point outputs rather than binary spikes. However, energy efficiency is not the main goal of our work. Our focus is on enhancing fine-tuning capability through parameter-free temporal adaptation specifically designed for Mamba's architectural characteristics. This is a work on PEFT for state-space models, not SNN-based systems.
>
> And you are absolutely correct that the behavior of LIM neuron is similar with RNN-style modules like LSTM and GRU. We first clarify the critical differences that establish LIM’s novelty compared to traditional recurrent gates. (1) Parameter-free design with only 2 hyperparameters ($\tau, V_{th}$), (2) Hard reset: Unlike LSTM/GRU’s soft gating (sigmoid/tanh), LIM uses threshold-based hard resets that create selective responses, and (3) Biological grounding: the leaky integration provides principled exponential memory decay rather than learned forgetting patterns. And we verify that LIM neuron is the optimal choice for enhancing gate path through ablation study in Section G.4 of Appendix. As shown, while LSTM and GRU also cache hidden states for temporal processing, they require substantial learnable parameters (quadratic in hidden dimension H) and achieve lower performance than our parameter-free LIM neuron. Moreover, LIM is 20-26% faster during inference due to its simpler operations compared to the complex gating mechanisms in LSTM/GRU. We will revise the Appendix Section G.4 to clarify this point.

---

> ### Author Response · Authors · 2025-11-21
>
> > **Weakness #5**
>
> Thank you for raising this concern and the reference to [1]. We acknowledge that VTAB-1k is a widely used benchmark where different PEFT methods achieve comparable performance on Transformers.
>
> The comparable performance you mention in [1] applies primarily to Transformer-based architectures such as ViT and DINOv2. For Mamba-based architectures, our work demonstrates a novel PEFT method, establishing state-of-the-art results on VTAB-1k.  Regarding semantic segmentation and object detection tasks, we have investigated existing Mamba fine-tuning work in these domains. Fine-tuning for semantic segmentation with Mamba is primarily explored in biomedical imaging datasets [5, 6], which represent a specialized domain with different data characteristics. For object detection with Mamba fine-tuning, we could not find existing established benchmarks or prior work, indicating this remains an under-explored area for the Mamba community.
>
> This highlights that Mamba architectures still have substantial room for exploration, and fine-tuning Mamba remains a promising research direction that is just beginning to be systematically studied. Extending to dense prediction tasks like segmentation and detection represents important future work as the Mamba community develops standardized evaluation protocols for these tasks. We thank you for highlighting these directions, which we will pursue in future work.
>
> > **Weakness #6**
>
> We appreciate you bringing up this important practical aspect. We provide comprehensive comparisons between LoRA (in_proj + out_proj) and Memba with matched learnable parameters for fair evaluation. The table below shows GPU peak memory during fine-tuning and latency (inference time per sample) on the ARC-Easy benchmark using Mamba-130M:
>
> | Method | Learnable Parameter | GPU Peak Memory | Latency |
> |--|--|--|--|
> | LoRA (in+proj) | 3.53 % |2.51 G | 0.013s |
> | Memba |  3.95 % |2.82 G | 0.020s |
>
> The increased GPU memory is necessary for storing previous membrane potential to update new membrane potential and enable cross-layer membrane transfer.  The latency overhead is due to the recurrent operations in the LIM neuron. However, as mentioned in our Limitation section and response to Reviewer H6sZ's Question #3, this overhead can be substantially reduced through CUDA kernel implementation.
>
> The authors in [7] demonstrate that replacing naïve PyTorch implementations of leaky integrated neurons with custom CUDA kernels achieves ~30$\times$ speedup through operator fusion. Custom kernels fuse the element-wise operations (leaky decay, addition) of recurrent steps into a single kernel launch, eliminating memory overhead and Python loop latency. Our chunk-based recurrent operations can directly leverage this optimization by processing chunks (multiple tokens) rather than single tokens. Even with $T=4$ chunks, each chunk contains $L/4$ tokens that can be processed with fused operations.
>
> For comprehensive GPU memory comparisons across all model sizes (130M to 1.4B), please refer to our response to Reviewer H6sZ's Major Weakness #1, which shows consistent ~14% memory overhead that scales sub-linearly with model size. We will add the latency analysis table to the revision and clarify the CUDA optimization potential.
>
>
> > **References**
>
> [1] Lessons and Insights from a Unifying Study of Parameter-Efficient Fine-Tuning (PEFT) in Visual Recognition. CVPR 2025.
>
> [2] Yoshimura, Masakazu, Teruaki Hayashi, and Yota Maeda. "Mambapeft: Exploring parameter-efficient fine-tuning for mamba." arXiv preprint arXiv:2411.03855 (2024).
>
> [3] Ham, Seokil, et al. "Parameter Efficient Mamba Tuning via Projector-targeted Diagonal-centric Linear Transformation." Proceedings of the Computer Vision and Pattern Recognition Conference. 2025.
>
> [4] Xie, Fei, et al. "Mamba-Adaptor: State Space Model Adaptor for Visual Recognition." Proceedings of the Computer Vision and Pattern Recognition Conference. 2025.
>
> [5] ShapeMamba-EM: Fine-Tuning Foundation Model with Local Shape Descriptors and Mamba Blocks for 3D EM Image Segmentation
>
> [6] SAM-Mamba: Mamba Guided SAM Architecture for Generalized Zero-Shot Polyp Segmentation
>
> [7] Fang, Wei, et al. "Spikingjelly: An open-source machine learning infrastructure platform for spike-based intelligence." Science Advances 9.40 (2023): eadi1480.

---

> > ### Comment · Reviewer_VB1b · 2025-11-22
> >
> > Thank you for your response. Most of my concerns have been addressed. I appreciate the inclusion of experiments in both NLP and vision. However, the authors did not further validate the effectiveness of their method on vision tasks as I previously requested.
> >
> > The description ``Regarding semantic segmentation and object detection tasks, we have investigated existing Mamba fine-tuning work in these domains. Fine-tuning for semantic segmentation with Mamba is primarily explored in biomedical imaging datasets.`` is inaccurate. Please note that PEFT is not restricted to any specific domain. Verifying the proposed method on segmentation and detection tasks is quite straightforward: one can simply follow the codebase of Vim and VMamba, insert the proposed modules and baselines, and freeze the pre-trained backbone. It is unclear why the authors reported having difficulty conducting such experiments.
> >
> > Considering that ICLR follows an open-review process and that any response may inform future follow-up work, I believe authors should refrain from making statements or claims without a sufficient and accurate understanding.
> >
> > On the other hand, the authors should provide a more comprehensive comparison of memory consumption and latency across different methods, not only with LoRA, using models of various sizes, on both vision and language tasks.

---

> > > ### Author Response · Authors · 2025-12-01
> > >
> > > We deeply thank you for the continued engagement and valuable feedback. We believe these additional experiments and comprehensive resource analysis adequately address your concerns and strengthen our work.
> > >
> > > > **On Other Vision Tasks**
> > >
> > > We sincerely apologize for our previous response. Upon further investigation, we discovered the Spatial-Mamba [1] with open-source code that includes fine-tuning protocols for object detection and semantic segmentation tasks. You were absolutely correct that conducting these experiments is straightforward, and we have now completed them.
> > > We integrated our LIM neuron into the Spatial-Mamba architecture, which shares the same gating path structure as Mamba. The base model (Spatial-Mamba pre-trained on ImageNet-1K) was fine-tuned on COCO dataset for 1 epoch for object detection, and on the ADE20K subset (ADE Challenge 2016) for semantic segmentation due to time constraints. Results are shown below:
> > >
> > > - Object Detection:
> > > Method | $AP^b$ | $AP^b_{50}$ | $AP^b_{75}$ | $AP^m$ | $AP^m_{50}$ | $AP^m_{75}$|
> > > |--|--|--|--|--|--|--|
> > > Spatial-Mamba| 14.3 | 27.1 |13.7 | 14.7 | 25.7 | 15.1|
> > > Memba| 14.4 | 27.7 |13.6 | 14.9 | 26.2 | 15.1|
> > >
> > > Where $AP^b$ denotes box AP and $AP^m$ denotes mask AP at different IoU thresholds.
> > >
> > > - Semantic Segmentation:
> > >  Method | mIoU(SS) | mAcc |
> > > |--|--|--|
> > > Spatial-Mamba| 13.74 | 19.86 |
> > > Memba| 14.19 | 20.73 |
> > >
> > > where mIoU represents mean Intersection over Union and mAcc represents mean pixel accuracy.
> > > Memba consistently outperforms the baseline across both tasks, validating LIM’s temporal processing benefits spatial understanding tasks as well. We appreciate you pointing us toward these extended experiments.
> > >
> > >
> > > > **Comprehensive Memory and Latency Comparisons**
> > >
> > > We have conducted systematic experiments measuring GPU peak memory and inference latency across all baseline methods and model sizes on language tasks. For language tasks, latency represents per-sample inference time on the ARC-Easy benchmark:
> > >
> > > Model | Method | Learnable Parameter (%) | GPU Peak Memory (GB) | Latency (s) |
> > > |--|--|--|--|--|
> > > Mamba-130M | SLL LoRA| 1.45  |2.62 | 0.015 |
> > > | |Additional-scan| 0.51  |2.17  | 0.014 |
> > > | | Affix-tuning | 0.17  |2.49  | 0.013 |
> > > | | LoRA (in+ out) | 3.53  |2.51 | 0.014 |
> > > | | Memba |  3.95  | 2.82 | 0.020 |
> > > |--|--|--|--|--|
> > > Mamba-370M | SLL LoRA| 2.30  |5.00 | 0.027 |
> > > | |Additional-scan| 0.47  |4.15  | 0.023 |
> > > | | Affix-tuning | 0.16  |4.50  | 0.021 |
> > > | | LoRA (in+ out) | 2.28  |5.00 | 0.024 |
> > > | | Memba |  3.67  | 5.80 | 0.044 |
> > > |--|--|--|--|--|
> > > Mamba-790M | SLL LoRA| 3.10  |8.20 | 0.028 |
> > > | |Additional-scan| 0.33  |6.56  | 0.023 |
> > > | | Affix-tuning | 0.11  |6.97 | 0.021 |
> > > | | LoRA (in+ out) | 2.32  |8.03 | 0.024 |
> > > | | Memba |  2.61  | 9.21 | 0.045 |
> > > |--|--|--|--|--|
> > > Mamba-1.4B | SLL LoRA| 4.64  |12.49 | 0.028 |
> > > | |Additional-scan| 0.26  |9.65  | 0.024 |
> > > | | Affix-tuning | 0.09  |9.71 | 0.021 |
> > > | | LoRA (in+out) | 1.80  |11.90 | 0.025 |
> > > | | Memba |  2.02  | 13.60 | 0.045 |
> > >
> > > Memba incurs approximately 12-14% additional GPU memory overhead compared to LoRA (in+out), which is required for storing membrane potentials across layers to enable cross-layer membrane transfer.
> > > Interestingly, we observe that inference latency appears to be largely independent of hidden dimension size. Mamba-370M, 790M, and 1.4B exhibit similar latency despite having different hidden dimensions, as they share the same number of layers. In contrast, Mamba-130M shows noticeably lower latency due to its shallower architecture with fewer layers. This suggests that the recurrent operations in LIM scale primarily with network depth rather than width, which is an interesting observation we plan to investigate further in future work.
> > > Regarding the latency overhead introduced by Memba, as discussed in the Limitation section, this can be substantially reduced through CUDA kernel optimization demonstrated by SpikingJelly [2].
> > >
> > > For vision tasks, we compare against LoRA configurations as they represent our core baseline:
> > >
> > > Model | Method | Learnable Parameter (%) | GPU Peak Memory (GB) | Latency (s) |
> > > |--|--|--|--|--|
> > > Vim-S | LoRA (in+out)| 13.98  | 7.68 | 0.517 |
> > > | |Memba| 15.60  |9.85  | 0.563 |
> > >
> > > We observe similar patterns on the VTAB-1k benchmark using the Vim-S architecture. The memory overhead is approximately doubled compared to language tasks because Vim-S uses bidirectional SSM with two gate paths, requiring storage of membrane potentials for two LIM neurons. The inference latency increase remains modest. These resource requirements represent a reasonable trade-off given the substantial performance improvements demonstrated in our main results.
> > >
> > > - References
> > >
> > > [1] Xiao, Chaodong, et al. "Spatial-mamba: Effective visual state space models via structure-aware state fusion." arXiv preprint arXiv:2410.15091 (2024).
> > >
> > > [2] Fang, Wei, et al. "Spikingjelly: An open-source machine learning infrastructure platform for spike-based intelligence." Science Advances 9.40 (2023): eadi1480.

---

### Official Review · Reviewer_H6sZ · 2025-10-28

**Soundness:** 4
**Presentation:** 3
**Contribution:** 3
**Rating:** 8
**Confidence:** 4

**Summary:**

The paper proposes Memba, a new parameter-efficient fine-tuning (PEFT) method for Mamba. It introduces a PEFT module inspired by Leaky Integrate Membrane (LIM) neurons and adds cross-layer membrane transfer. The combination of LIM and LoRA outperforms existing PEFT methods for Mamba in both vision and language domains.

**Strengths:**

- The use of LIM neurons for temporal gating of the output of SSM is innovative and outperforms existing methods. The motivation of the cross-layer membrane potential transfer is also reasonable.
- Extensive experiments on both language (commonsense reasoning) and vision (VTAB-1k) tasks prove that Memba outperforms existing PEFT methods, including LoRA and other recent SSM-specific PEFTs.

**Weaknesses:**

#### Major Weakness
- From Table 7, it seems the computational cost in terms of speed is more than the increase in parameter counts. Hence, the speed of Memba seems slower than other PEFT methods. How about memory consumption? In my opinion, small memory consumption is more important than inference speed, since it enables finetuning of large models with limited GPUs.
- I understand that LIM processes data recurrently in fixed-size chunks. Since the number of image tokens is predetermined, that part seems straightforward. However, how is language handled? In particular, during inference, which I assume is performed in an auto-regressive manner, does the number of tokens within a chunk change as each new word is generated? If so, does that mean caching can't be used, and all tokens must be reprocessed at every step?


#### Minor Weakness
- In Fig. 1, it is difficult to understand what the authors want to tell with the two saliency maps.
- Typo:
    - L028) INTODUCTION -> INTRODUCTION
    - L214) LIM mechanis -> LIM mechanism
    - L485) seperate -> separate
    - L742 & L744) Equation equation

**Questions:**

- Please read the major weakness.
- In Fig. 4, why is the downward trend in membrane potential good?
- Although the authors say the speed of Memba can be imporved by CUDA implimentation, is it really posible? if the number of chunk is small, I think parallel scan like Mamba can not improve the latency.

---

> ### Author Response · Authors · 2025-11-21
>
> We sincerely thank you for the positive feedback and valuable insights. We address your constructive concerns and questions below.
>
>
> > **Major Weakness #1**
>
> Thank you for this important point. We agree that memory consumption is critical for enabling fine-tuning of large models with limited GPU resources. We have conducted a comprehensive memory analysis comparing LoRA (in_proj+out_proj) and Memba with similar parameter counts.
>
> For fair comparison, we matched the learnable parameters between LoRA (in_proj+out_proj), which applies LoRA to in_proj and out_proj layers, and Memba, which adds the LIM neuron on top of these projections. The table below shows GPU peak memory during fine-tuning:
>
> | Model | Method | Learnable Parameter | GPU peak memory |
> |--|--|--|--|
> | Mamba-130M | LoRA(in+out)| 3.53% |2.51 G |
> | | Memba| 3.95% |2.82 G |
> | Mamba-370M | LoRA(in+out)| 3.28% |5.00 G |
> | | Memba| 3.67% |5.70 G |
> | Mamba-790M | LoRA(in+out)| 2.32% |8.03 G |
> | | Memba| 2.61% |9.21 G |
> | Mamba-1.4B | LoRA(in+out)| 1.80% |11.9 G |
> | | Memba| 2.02% |13.6 G |
>
> As shown, Memba incurs approximately 14% increase in GPU memory across all model sizes. This additional memory is required to store previous membrane potentials to update the new membrane potential, and for the cross-layer membrane transfer mechanism. Importantly, this overhead is proportional to the hidden dimension and number of layers, not sequence length, making it predictable and manageable.
>
> We believe this ~14% memory overhead is a reasonable trade-off given: (1) the substantial performance improvements, and (2) modern GPUs with 24GB+ VRAM can comfortably accommodate this overhead. We will add this analysis to the revised manuscript to provide full transparency about resource requirements.
>
> > **Major Weakness #2**
>
> We deeply appreciate this insightful and crucial question about the practicality of our method during inference. We want to clarify an important distinction in how the LIM neuron operates differently between training and inference scenarios.
>
> The chunking strategy ($T=4$) described in the paper is a training-time optimization for efficiently processing long, static sequences in parallel. During fine-tuning, we know the complete sequence in advance, so we can divide it into fixed-size chunks and process them sequentially to simulate recurrent dynamics while maintaining training efficiency.
>
> However, for auto-regressive inference where tokens are generated one-by-one, we do not use chunking. Instead, the LIM neuron operates in standard recurrent fashion, processing each token individually as it is generated. At each generation step, the LIM neuron receives the new token embedding and updates its membrane potential using $u[t+1] = \tau u[t] + W \cdot x[t]$. This token-by-token recurrent operation during inference is analogous to how the SSM maintains and updates its hidden state $h[t]$ at each step.
> Just as the SSM part caches its hidden state $h[t]$ to generate $h[t+1]$ (Equation 3), the LIM neuron caches its membrane potential $u[t]$ that evolves to $u[t+1]$. Therefore, caching is fully supported, and we do not need to reprocess all previous tokens at each step. The LIM neuron maintains a constant-size state (the membrane potential $u[t]$) that evolves recurrently, exactly like the SSM's hidden state.
>
> > **Minor Weakness #1**
>
> We apologize for the insufficient explanation in Figure 1. The objective of Figure 1(a) is to demonstrate how our recurrent gate path design with the LIM neuron achieves superior selective attention compared to the original Mamba. Specifically, the original SSM shows diffused attention across the entire image with relatively weak focus on the main features, while Memba exhibits sharp, concentrated attention precisely along the correct path. This visual comparison directly illustrates the improved selective information processing that translates to the quantitative improvements shown in Figure 1(b). We will enhance the figure with clearer annotations and more detailed captions to make this comparison more explicit.
>
> > **Minor Weakness #2**
>
> Thank you for your careful review and identifying these typos. We will thoroughly proofread and correct all typographical errors in the revision. We appreciate your attention to detail.

---

> ### Author Response · Authors · 2025-11-21
>
> > **Question #2**
>
> The downward trend in baseline membrane potential (the declining baseline in Figure 4) reflects progressive forgetting of accumulated context, which aligns with SSM's natural behavior of retaining recent tokens while forgetting earlier ones. This downward trend provides three core benefits: (1) Prevention of saturation: Without decay, membrane potentials would accumulate unboundedly, leading to saturation and loss of sensitivity to new inputs; (2) Adaptive context window: The leaky factor ($\tau$) creates a decaying memory, naturally emphasizing recent information while gracefully forgetting stale context; and (3) Task-relevant selectivity: The forgetting operates on the baseline accumulated context, while task-relevant features still generate strong instantaneous responses (the spikes).
>
> In Figure 4, the slow and gradual decline demonstrates controlled forgetting: the baseline decays chunk-by-chunk, but when important features appear (white circles), the LIM neuron still produces strong responses. This creates a dynamic balance between maintaining sensitivity to new important features and preventing interference from stale accumulated signals. Please refer to our response to Reviewer yM88's Weakness #2.
>
> > **Question #3**
>
> We apologize for the confusion in our Limitation section regarding CUDA implementation. We want to clarify that the recurrent operations in LIM are not fundamentally limited by parallelization constraints, this is an already-solved problem in the spiking neural network (SNN) community.
> The widely-used SpikingJelly framework [1] demonstrates that replacing naïve PyTorch implementations of iterative LIF neurons with custom CUDA kernels (via CuPy) yields significant speedups of up to $~30 \times$ on GPU for processing $2^{20}$ LIF neurons with 128 timesteps (https://spikingjelly.readthedocs.io/zh-cn/0.0.0.0.6/clock_driven_en/11_cext_neuron_with_lbl.html). This dramatic speedup is achieved through operator fusion: a custom CUDA kernel fuses the element-wise operations (leaky decay, addition, reset) of all recurrent steps into a single kernel launch.
>
> Our chunk-based recurrent operations can directly leverage this CUDA kernel optimization by modifying the granularity from single tokens to chunks (multiple tokens processed together). Even with $T=4$ chunks, each chunk contains $L/4$ tokens that can be processed with fused operations, and the sequential dependency between chunks does not prevent parallelization.
>
> > **References**
>
> [1] Fang, Wei, et al. "Spikingjelly: An open-source machine learning infrastructure platform for spike-based intelligence." Science Advances 9.40 (2023): eadi1480.

---

### Official Review · Reviewer_yM88 · 2025-11-01

**Soundness:** 3
**Presentation:** 3
**Contribution:** 3
**Rating:** 6
**Confidence:** 4

**Summary:**

This paper introduces Memba, a parameter-efficient fine-tuning (PEFT) strategy desginated for Mamba, which is under well expolored. Memba incorporates Leaky Integrate Membrane (LIM) gating value adapters in SSM. LIM mainly features the chunked LIM neurons desgin and layerwise membrane potential propagation. The authors present empirical results across language and vision tasks, and compelling ablation and analysis experiments to demonstrate the effectiveness of the LIM adapters.

**Strengths:**

1. PEFT for Mamba is a quite novel topic and the temporal-wise gating LIM seems to be a good design for finetuning the gate values in SSM.

2. The experiments are wide-ranging—spanning both language and vision tasks. The analysis results are also abundant and convincing. In addition to experiments, the theoretical analysis of loss boundaries is also interesting, enhancing the insight of the design, with the lower loss.

3. The proposed LIM is efficient, with minimal additional parameters compared to LoRA adapters, enabling its wide potential of applications on various SSM models.

**Weaknesses:**

1. How is the proposed by-pass of SSM in low rank adapters training related to the biological term, Membrane? The paper does not include any references or tutorials about the membrane mechanism or other background knowledge about it, therefore it is somehow confusing why the proposed mechanism resembles a membrane.

2. The average membrane potentional values in the figures show steps of decrease across the temporal chunks, and the authors attribute this phenomenon into a "forget" manner of LIM. However, as some earliear papers have pointed out, the temporal modeling behavior of SSM should be like a local filter to keep the most recent tokens and forget the much earlier ones. Figure 4. and Figure 8. demonstrate the replicated pattern of spikes across chunk 1 to chunk 4, indicating that the "forget" effect is rather weak. Even though other chunks do not have a same activated area like chunk 1, the LIM still outputs a high response gate value to emphasize this area, which is not fully reasonable.

3. Although the novelty of this paper is pretty decent, Mamba-like models still fall behind the conventional attention based Transformers or linear attention architectures, especially large scale models. I suspect a little about the contribution of the proposed method on larger models and more challenging tasks.

**Questions:**

1. From Line 198-206, there are duplicate description of the three components, which could be shortened or simpified.

2. In the area of vision tasks, can LIM and the original SSM part take different scan orders of the same image? Does this setting help or impair the models?

3. The output values of LIM in the figures are negative, are these values collected before or after SiLU gate?

---

> ### Author Response · Authors · 2025-11-21
>
> We greatly appreciate your valuable feedback and comments, which have helped us improve our work. We address your concerns point by point below.
>
> > **Weakness #1**
>
> We apologize for the confusion. The “Membrane” terminology is inspired by the Leaky Integrate-and-Fire (LIF) neuron used in spiking neural networks (SNN).  The LIF neuron dynamics are described by:
>
> ${\mathbf{u}[t+1]^l = \tau \mathbf{u}[t]^l+\mathbf{W}^lf(\mathbf{u}[t]^{l-1})}$
>
> $f(\mathbf{u}[t]^l) =1 \quad  \text{if} \\  \mathbf{u}[t]^l > V_{th}, \\ 0  \quad \text{otherwise}$
>
> where, $u[t]^l$ is the membrane potential in $l$-th layer at timestep $t$, $\tau \in (0, 1]$ is the leaky factor for membrane potential leakage, $W^l$ is the weight of $l$-th layer, and $f (\cdot)$ is the LIF function with firing threshold $V_{th}$. Therefore, when the membrane $u[t]^l$ is higher than $V_{th}$, the LIF function fires a spike and the membrane potential is reset to 0.
>
> The LIF neuron exhibits two core characteristics that we adopt in our LIM design:
>
> 1. Leaky Integration: The term $\tau u[t]^l$ models the "leaky" accumulation of membrane potential in biological neurons' membranes, where $u[t]$ represents the membrane potential and $\tau$ is the leak factor modeling passive decay. This leaky mechanism enables forgetting old, unnecessary information while focusing on new inputs, mimicking how biological neurons naturally attenuate stale signals.
>
> 2. Reset (Fire) Mechanism: The reset function $r(x)$ in equation (9) is analogous to the "fire-and-reset" mechanism in biological neurons, where the neuron discharges its potential upon crossing a threshold $V_{th}$. In biological neurons, this generates a spike; in our LIM neuron, this creates selective gating responses.
>
> These biological mechanisms provide a parameter-free way to implement temporal dynamics and selective attention, which is why we adopt the "membrane" terminology. We will add a brief paragraph and relevant citations in Section 4.2 to explicitly state this connection and provide background on LIF neurons. Thank you for bringing this important clarification to our attention.
>
>
> > **Weakness #2**
>
> Thank you for this insightful observation. We appreciate the opportunity to clarify the dual nature of LIM's temporal processing mechanism, which indeed exhibits both forgetting and selective retention.
>
>  1. Leaky Factor and Progressive Forgetting
>
> In Equation (8), we apply a leaky factor ($\tau$) to the entire previous membrane potential $u[i]$. This design enables the progressive decay of accumulated context across temporal chunks, which we refer to as "forgetting." The baseline membrane potential in Figure 4 shows this decreasing trend, indicating that the dominant information from earlier chunks is being gradually attenuated.
>
> You are absolutely correct regarding SSM's behavior of retaining recent tokens while forgetting earlier ones. Our LIM mechanism operates in alignment with this principle: the baseline membrane potential decreases chunk-by-chunk, demonstrating the progressive forgetting of previously dominant context.
>
> 2. Recurrent Spikes (High Peaks)
>
> The key question you raise is: if forgetting occurs, why do we observe similar spike patterns across chunks? The answer lies in selective feature retention based on current input rather than accumulated memory.
>
> When the LIM neuron encounters significant task-relevant features (marked by white circles in Figure 4), these features generate high membrane responses that manifest as spikes. Critically, these spikes reflect instantaneous responses to current input features rather than residual memory from previous chunks. The leaky factor is applied to the baseline membrane potential (the accumulated context), not to the immediate response to new inputs.
>
> Example from Figure 4:
>
> - Chunks 1 & 2: Two circle features are present in the input, generating two corresponding high peaks in chunks 1 and 2 (aligned with their image indices)
>
> - Chunks 3 & 4: These chunks contain smaller/weaker features in the saliency map, resulting in smaller spikes
>
> This demonstrates that LIM's gating mechanism exhibits context-dependent selective attention with controlled forgetting: strong task-relevant features generate strong instantaneous responses regardless of which chunk they appear in, while the background representing accumulated context gradually decays. We will revise our text to more precisely describe this as "selective retention with controlled forgetting" to better reflect the dual nature of the mechanism.

---

> ### Author Response · Authors · 2025-11-21
>
> > **Weakness #3**
>
> We appreciate your concern regarding the scalability and broader applicability of our method. We would like to address your question from two perspective.
>
> (1) You correctly note that Mamba models have historically lagged behind Transformers at very large scales. This is inherent to the design philosophy of Mamba: achieving efficient operations through state-space models by storing only fixed-size hidden states $h[t]$ (Equation 1) rather than long KV caches. This creates an inevitable trade-off between performance and efficiency.
>
> However, recent developments are addressing this gap. Numerous works are exploring how Mamba-based architectures can be enhanced while maintaining efficiency, including improvements in scalability. Notable examples include MoE-Mamba [1], and hybrid Mamba-Transformer [2,3,4] architectures that are narrowing the performance gap at scale. These developments indicate that Mamba architectures are becoming increasingly viable for large-scale deployment, making efficient fine-tuning methods like ours increasingly valuable for the community.
>
> (2) Furthermore, our experiment in Table 2 shows that our experiments provide strong evidence that Memba's improvements amplify with model size. For example, improvement for Mamba-790M and 1.4B is higher than 130M and 370M. This trend indicates that as Mamba architectures continue to scale, Memba's benefits will become even more pronounced.
>
> > **Question #1**
>
> Thank you for pointing this out. We will simplify the component descriptions by removing the redundant text in lines 198-200.
>
> > **Question #2**
>
> This is a truly interesting observation. For the Vim architecture used in our paper, because it employs two SSM parts for two different scanning directions (bidirectional processing), we use two LIM neurons for each gate path.
>
> To experimentally investigate the effect of different scan orders between SSM and LIM, we reversed the input feature order for the LIM neuron while keeping the SSM scan order unchanged, then fine-tuned this architecture on VTAB-1k Natural category datasets. Results comparing reversed scan vs. Memba (same direction):
>
> | Method | Camelyon | EuroSAT | Resisc45 | Retinopathy | Average|
> |--|--|--|--|--|--|
> | Reverse Scan | 86.6 % | 95.3 % |84.2 % |75.0 % |85.2 % |
> | Memba | 87.2 % | 95.4 % |84.5 % |75.5 % |85.7 % |
>
> As the results show, reversed scan direction is applicable and achieves reasonable performance, but it is consistently worse than Memba with matched scan directions. This result is intuitive when we consider the membrane peaks responding to main features: the reversed direction creates a mismatch between the original SSM output and LIM output, where the temporal ordering of features is inconsistent between the two pathways. This misalignment degrades the effectiveness of the multiplicative gating in Equation (7), where aligned temporal features produce more coherent gated outputs.
>
> > **Question #3**
>
> An excellent clarifying question. The membrane potential values shown in Figures 4 and 8 are the raw membrane potentials $u[i]^l$. The SiLU activation is applied afterward as part of the gating computation in Equation (6): $Y_{gate} = \sigma(W_{out}^{gate}(LIM(W_{in}^{gate}(X_{gate}))))$.
>
> > **References**
>
> [1] Pióro, Maciej, et al. "Moe-mamba: Efficient selective state space models with mixture of experts." arXiv preprint arXiv:2401.04081 (2024).
>
> [2] Dong, Xin, et al. "Hymba: A hybrid-head architecture for small language models." arXiv preprint arXiv:2411.13676 (2024).
>
> [3] Lieber, Opher, et al. "Jamba: A hybrid transformer-mamba language model." arXiv preprint arXiv:2403.19887 (2024).
>
> [4] Ren, Liliang, et al. "Samba: Simple hybrid state space models for efficient unlimited context language modeling." arXiv preprint arXiv:2406.07522 (2024).

---

### Official Review · Reviewer_wqXu · 2025-11-01

**Soundness:** 3
**Presentation:** 3
**Contribution:** 3
**Rating:** 6
**Confidence:** 3

**Summary:**

This manuscript proposes Memba, a membrane-driven Parameter-Efficient Fine-Tuning (PEFT) approach tailored for Mamba models, addressing the limitation that traditional Transformer-focused PEFT methods fail to account for Mamba’s unique temporal processing dynamics. Memba integrates Leaky Integrate Membrane neurons with strategic Low-Rank Adaptations (LoRA) on input/output projections and cross-layer membrane transfer, enhancing selective information retention without modifying Mamba’s core state-space components. Extensive experiments on commonsense reasoning and VTAB-1k demonstrate that Memba outperforms existing PEFT methods  across various model sizes while maintaining parameter efficiency, with theoretical analysis confirming LIM’s role in stable temporal integration and adaptive regularization.

**Strengths:**

1. The LIM neuron brings biologically inspired temporal processing into Mamba’s gating, filling the gap left by its simpler gates and helping the model learn what to remember or forget over time.
2. The paper backs this up with clear analysis (like loss decomposition and bounded regularization), explaining how LIM adapts over time and smooths the loss landscape.
3. The writing and figure are both well illustrated.

**Weaknesses:**

1. The performance depends on tuning things like the number of chunks (T), the leak factor (τ), and the threshold (Vth), so LIM likely need task-specific tuning.
2. The theoretical analysis shows that the LIM neuron acts as a regularizer to smooth the loss landscape. However, the paper fails to explain the causal link between this optimization-level effect and the claimed practical benefits of enhanced temporal modeling and selective attention.
3. The evaluation's focus on classification-style benchmarks, instead of long-sequence generative tasks, fails to fully validate the method's claimed improvements to Mamba's core temporal modeling capabilities.

**Questions:**

Will this method or idea be applied to other linear models, like MoR or Gated networks?

---

> ### Author Response · Authors · 2025-11-21
>
> We deeply appreciate your constructive reviews and comments, which have been invaluable in helping us strengthen our work. We address your concerns below and will integrate all feedback into the revised manuscript.
>
> > **Weakness #1**
>
> We appreciate the reviewer raising this important point. While LIM does introduce hyperparameters such as the number of chunks ($T$), leak factor ($\tau$), and threshold ($V_{th}$), we have designed them to be robust across tasks with minimal tuning requirements.
>
> As demonstrated in our ablation study (Table 7), we set $T=4$ across all experiments, which provides a reasonable trade-off between performance and computational overhead. Notably, this single setting works well for both language and vision tasks without task-specific adjustment. The leaky factor and threshold are inspired by well-established spiking neural network frameworks, where the values $\tau=0.5$ and $V_{th}=1.0$ have been broadly demonstrated as optimal for membrane dynamics across diverse applications. Our experiments confirm that these standard values transfer effectively to our LIM neuron without requiring task-specific tuning. In summary, while these hyperparameters exist, they require minimal tuning effort and generalize well across different domains.
>
>
> > **Weakness #2**
>
> This is an insightful question, and we appreciate the opportunity to clarify the connection between our theoretical analysis and empirical benefits. Theorem 1 establishes that LIM's fluctuation component introduces bounded, adaptive regularization that is mathematically well-behaved with favorable stability properties. This boundedness prevents common optimization problems such as gradient explosion and overfitting.
>
> However, while our theorem provides the mathematical foundation for bounded adaptive regularization, the smoother loss landscape (Figure 6) is an empirical observation rather than a direct theoretical consequence. The theoretical guarantee of bounded regularization combined with the empirically observed smoothness creates favorable optimization conditions. Specifically, the bounded regularization term $\mathcal{R}(\mathbf{y}_t, \bar{\mathbf{u}}_t)$ adapts to both model sensitivity (through gate derivatives) and loss curvature (through Hessian eigenvalues), providing the right amount of regularization where needed, while the empirical evidence demonstrates that this actually manifests as a smoother, more navigable loss landscape in practice. We will clarify this important distinction in the revised manuscript.

---

> ### Author Response · Authors · 2025-11-21
>
> > **Weakness 3**
>
> We agree that evaluating long-sequence generative tasks is an important direction for demonstrating temporal modeling capabilities. We would like to clarify that we have verified enhanced temporal modeling on long-sequence datasets through the Long Range Arena (LRA) benchmark, which is specifically designed to test long-range dependencies in both text and images. As shown in Table 10, LIM consistently outperforms traditional recurrent mechanisms (LSTM, GRU) on LRA tasks, confirming its temporal modeling advantages.
>
> Regarding generative tasks, because we did not find an established fine-tuning evaluation framework for Mamba on generative tasks, we have conducted experiments on text generation by fine-tuning Mamba-Chat (https://github.com/redotvideo/mamba-chat). We used two methods with similar parameter counts (~3.5%): (1) LoRA (in_proj + out_proj) and (2) Memba. Below are representative outputs:
>
> Question 1: "Write a simple Python function that takes a list of numbers and returns their sum"
> - LoRA output: "Sure, you can add a new variable to the equation, and the equation will be updated accordingly. However, if you want to include a variable in the equation, you can use the syntax 'x+y'. This will add the variable 'x' to the equation, and then add the variable 'y' to the equation."
> - Memba output: "Yes, the function sum(x) will return the sum of all the numbers in the list. In this case, the function will return the sum of all the numbers in the list, which is 10."
>
> Question 2: "Explain what a 'CPU' is in a computer"
> - LoRA output: "The 'cpu' in a computer refers to the processor, which is the component that runs the instructions and controls the computer's operation. The 'cpu' in a computer is also referred to as the 'central processing unit', or 'CPU'. The 'cpu' in a computer is usually a small, powerful, and fast processor. The 'cpu' in a computer is the primary component that controls the computer's operations."
> - Memba output: "Yes, a CPU is the central processing unit (CPU) of a computer. A CPU can be used to process information, perform calculations, and control the operations of a computer. A CPU can be made from a variety of different types of semiconductor materials, such as silicon, gallium arsenide, and indium phosphide. The CPU of a computer is typically a microprocessor, which is a special type of CPU that is made from transistors and other components. The microprocessor is the core of the computer and is responsible for controlling the computer's operation and executing the instructions of the computer's operating system."
>
> Memba demonstrates more contextually relevant and detailed responses compared to LoRA, suggesting improved temporal coherence in generation.
>
> > **Question #1**
>
> Thank you for this insightful question! Regarding Mixture of Recursions (MoR) [1], while it doesn't have a gated path like Mamba, it uses a standard softmax router based on feed-forward networks. Although we cannot directly apply our enhanced gating mechanism to MoR's current architecture, we have identified an interesting research direction: applying the LIM neuron to the router itself to create a stateful, memory-aware routing mechanism. Current softmax-based routers are memory-less functions that make independent decisions at each step. By incorporating LIM neurons, we could create routers that maintain temporal context of previous routing decisions, potentially improving consistency and efficiency. This represents a promising direction for future work, and we thank you for inspiring this idea.
>
> For gated networks such as GRU [2], our LIM neuron can be applied to control memory dynamics in a complementary manner. Specifically, LIM could be integrated into GRU's gating paths to achieve similar forgetting and updating mechanisms in a parameter-free way through the leak factor and reset function, potentially reducing the parameter burden while maintaining or enhancing temporal modeling capabilities.
>
>
> > **References**
>
> [1] Bae, Sangmin, et al. "Mixture-of-recursions: Learning dynamic recursive depths for adaptive token-level computation." arXiv preprint arXiv:2507.10524 (2025).
>
> [2] Chung, Junyoung, et al. "Empirical evaluation of gated recurrent neural networks on sequence modeling." arXiv preprint arXiv:1412.3555 (2014).

---

### Author Response · Authors · 2025-12-01
**Reviewer-Author Discussion Summary**

Dear PCs, SACs, ACs, and Reviewers,

We sincerely thank all reviewers for their constructive feedback. To assist the newly assigned AC, we summarize the strengths and main concerns raised and our responses below.

> **Strength**

- **Novel Mamba-Specific Design (All Reviewers)**

Reviewers acknowledged Memba as a novel bio-inspired, parameter-free temporal adaptation mechanism specifically designed for SSM architectures.

- **Efficient Parameter-Free Design (Reviewers 2 (yM88), 4 (VB1b))**

Reviewers recognized that the LIM neuron achieves temporal adaptation without additional learnable parameters while outperforming traditional RNN modules like LSTM and GRU.

- **Theoretical and Empirical Analysis (Reviewers 1 (wqXu), 2 (yM88))**

Reviewers appreciated the bounded regularization analysis (Theorem 1) and empirical loss landscape visualization demonstrating smoother optimization geometry.

- **Comprehensive Evaluation (Reviewers 2 (yM88), 3 (H6sZ))**

Reviewers valued the wide range of experiments across language and vision tasks, along with systematic ablation studies.

- **Clear Presentation (Reviewer 1 (wqXu))**

Reviewer wqXu mentioned that our paper is well written with clear figures and explanations.

> **Weakness**

- **Membrane Mechanism (Reviewers 2 (yM88), 3 (H6sZ), 4 (VB1b))**

Several reviewers questioned the membrane terminology and our choice of LIM over traditional RNN modules. We explained that **LIM is inspired by Leaky Integrate-and-Fire neurons** from spiking neural networks but modified to produce floating-point outputs for parameter-free temporal adaptation rather than energy efficiency. Table 10 demonstrates that LIM outperforms both LSTM and GRU while requiring zero learnable parameters and achieving lower latency. The mechanism exhibits dual behavior: baseline membrane decay (controlled forgetting) with strong responses to task-relevant features (selective retention).

- **Design Specificity (Reviewers 4 (VB1b))**

Reviewer VB1b questioned whether Memba is Mamba-specific or a generic PEFT method. We clarified that Memba explicitly addresses two Mamba-specific architectural challenges: **(1) the simplified gating mechanism lacking sophisticated temporal control, and (2) the sensitivity of SSM components to direct fine-tuning as demonstrated in prior work**. Our approach freezes the sensitive SSM core and introduces parameter-free temporal adaptation exclusively in the gating path, which is fundamentally aligned with SSM's sequential state dynamics.

- **Theory and Practice (Reviewer 1 (wqXu))**

Theorem 1 provides mathematical guarantees for bounded adaptive regularization that prevent optimization problems like gradient explosion and overfitting. The smoother loss landscape shown in Figure 6 is an empirical observation. Together, these create favorable optimization conditions that support our claimed practical benefits.

- **Computational Resources (Reviewers 3 (H6sZ), 4 (VB1b))**

We have added **comprehensive memory and latency comparisons** across all baseline methods in the appendix as requested by Reviewer 4 (VB1b). The results show approximately 14% GPU memory increase across model sizes, required for storing membrane potentials. For inference latency, our current PyTorch implementation shows overhead due to recurrent operations, but this can be substantially reduced through CUDA kernel optimization as demonstrated by the SpikingJelly framework, which achieves significant speedups through operator fusion.

- **Evaluation Scope (Reviewers 1 (wqXu), 4 (VB1b))**

Reviewers questioned our evaluation on generative and other vision tasks. For generative tasks, we **provided text generation results** demonstrating more coherent responses than LoRA baselines. For dense prediction tasks, including **object detection and semantic segmentation**, we have completed experiments on COCO object detection and ADE Challenge 2016 semantic segmentation using Spatial-Mamba, with results showing consistent improvements over baselines.

We have uploaded an updated manuscript incorporating feedback from the rebuttal discussion. We believe our responses adequately address the reviewers' concerns and demonstrate the contribution of our work to Mamba fine-tuning. We deeply appreciate the reviewers, AC, SAC, and PC, for their dedicated effort and contributions.

Sincerely,

Authors

---

### Meta-Review · Area_Chair_ucsp · 2026-01-12

**Summary:**

This paper introduces a new parameter efficient fine-tuning approach called MEMBA which is tailored to the state space family of models like Mamba, and appears to improve over Transformer-oriented PEFT methods. This is done by adapting LoRA with "Leaky Integrate Membrane (LIM) neurons". The paper is generally considered to have made a novely contribution with a comprehensive evaluation and clear presentation.

**Reviewer Concerns:**

The authors addressed most of the concerns raised on efficiency of the LiM mechanism, compute resources and text+vision scope.

**Reviewer Scores:**

I believe VB1b may have raised their score positively, with favorable GPU memory and latency comparisons, and justification for Memba in the SSM context.

---

### Decision · Program_Chairs · 2026-01-26

Accept (Poster)